# Glial ferritin maintains neural stem cells via transporting iron required for self-renewal in *Drosophila*

Zhixin Ma[1], Wenshu Wang[1], Xiaojing Yang[1], Menglong Rui[1], Su Wang[1,2]*

[1]School of Life Science and Technology, Department of Neurosurgery, Zhongda Hospital, The Key Laboratory of Developmental Genes and Human Disease, Ministry of Education, Southeast University, Nanjing, China; [2]Co-innovation Center of Neuroregeneration, Nantong University, Nantong, China

**Abstract** Stem cell niche is critical for regulating the behavior of stem cells. *Drosophila* neural stem cells (Neuroblasts, NBs) are encased by glial niche cells closely, but it still remains unclear whether glial niche cells can regulate the self-renewal and differentiation of NBs. Here, we show that ferritin produced by glia, cooperates with Zip13 to transport iron into NBs for the energy production, which is essential to the self-renewal and proliferation of NBs. The knockdown of glial ferritin encoding genes causes energy shortage in NBs via downregulating aconitase activity and NAD$^+$ level, which leads to the low proliferation and premature differentiation of NBs mediated by Prospero entering nuclei. More importantly, ferritin is a potential target for tumor suppression. In addition, the level of glial ferritin production is affected by the status of NBs, establishing a bicellular iron homeostasis. In this study, we demonstrate that glial cells are indispensable to maintain the self-renewal of NBs, unveiling a novel role of the NB glial niche during brain development.

*For correspondence:
wangsu@seu.edu.cn

**Competing interest:** The authors declare that no competing interests exist.

## eLife assessment

This **valuable** study, which seeks to identify factors from the glial niche that support and maintain neural stem cells, reports a novel role for ferritin in this process. The authors provide **solid** evidence that defects in larval brain development in Drosophila, resulting from ferritin knockdown, can be attributed to impaired Fe-S cluster activity and ATP production. The findings of this well-conducted study will be of interest to oncologists and neurobiologists.

## Introduction

Neural stem cells (NSCs) possess the remarkable ability to self-renew for maintaining their identity and differentiate into diverse neural cell types required for neurogenesis during brain development. The self-renewal, differentiation, and proliferation of NSCs are regulated by both intrinsic programs inside stem cells and extrinsic cues from the surrounding niches. The mammalian NSC niche found in the neurogenic regions is composed of diverse cell types including choroid plexus, vascular endothelia cells, pericytes, microglia, and meninges, which regulate NSC behaviors such as proliferation, self-renewal, and differentiation (*Bjornsson et al., 2015*). The diversity and complexity of these cell populations confer niche the ability to integrate local and systemic signals, but make it complicated to study the interaction between niches and NSCs in mammals, which is a critical issue in understanding neurogenesis and improving clinical applications of NSCs.

*Drosophila* larval central nervous system (CNS), composed of the central brain (CB), optic lobe (OL), and ventral nerve cord (VNC), is a simple but sophisticated model to decipher the regulatory

**eLife digest** Iron is an essential nutrient for almost all living organisms. For example, iron contributes to the replication of DNA, the generation of energy inside cells, and the transport of oxygen around the body. Iron deficiency is the most common of all nutrient deficiencies, affecting over 40% of children worldwide. This can lead to anemia and also impair how the brain and nervous system develop, potentially resulting in long-lasting cognitive damage, even after the deficiency has been treated.

It is poorly understood how iron contributes to the development of the brain and nervous system. In particular, whether and how it supports nerve stem cells (or NSCs for short) which give rise to the various neural types in the mature brain.

To investigate, Ma et al. experimentally reduced the levels of ferritin (a protein which stores iron) in the developing brains of fruit fly larvae. This reduction in ferritin led to lower numbers of NSCs and a smaller brain. Unexpectedly, this effect was largest when ferritin levels were reduced in glial cells which support and send signals to NSCs, rather than in the stem cells themselves.

Ma et al. then used fluorescence microscopy to confirm that glial cells make and contain a lot of ferritin which can be transported to NSCs. Adding iron supplements to the diet of flies lacking ferritin did not lead to normal numbers of stem cells in the brains of the developing fruit flies, whereas adding compounds that reduce the amount of iron led to lower numbers of stem cells. Together, this suggests that ferritin transports iron from glial cells to the NSCs. Without ferritin and iron, the NSCs could not produce enough energy to divide and make new stem cells. This caused the NSCs to lose the characteristics of stem cells and prematurely turn into other types of neurons or glial cells.

Together, these findings show that when iron cannot move from glial cells to NSCs this leads to defects in brain development. Future experiments will have to test whether a similar transport of iron from supporting cells to NSCs also occurs in the developing brains of mammals, and whether this mechanism applies to stem cells in other parts of the body.

mechanisms underlying NSC behaviors mediated by niche signals. In this system, *Drosophila* NSCs, called NBs, and their progenies are encased by a variety of glial cells including perineurial glia, subperineurial glia, cortex glia, astrocyte-like glia and ensheathing glia (*Hartenstein, 2011*; *Ito et al., 1995*; *Pereanu et al., 2005*). These glial cells not only form the physical support structure for NBs, but also provide extrinsic cues that regulate NB reactivation, survival, and proliferation (*Read, 2018*; *Spéder and Brand, 2014*). However, it remains elusive whether the glial-cell-forming niche is involved in the regulation of the prominent characteristics of NBs, self-renewal, and differentiation.

Iron is a vital micronutrient for almost all living organisms since it participates in many crucial biological processes, including mitochondrial electron transport and cellular respiration that both support ATP production used for most cellular activities. High levels of iron are found in many organs including the brain, where iron is required for myelination and formation of neuronal dendritic trees (*Lozoff et al., 2006*; *Möller et al., 2019*; *Rice and Barone, 2000*). Iron deficiency in infants and early childhood has been well-documented to be associated with long-term cognitive, social, and emotional problems (*Fretham et al., 2011*). Thus, it is intriguing to investigate whether iron is also required in NSCs during neurogenesis.

Ferritin, a ubiquitous protein present in most organisms, is composed of 24 subunits that assemble into a hollow-sphere complex capable of storing up to 4500 iron atoms in a bioavailable form (*Andrews, 2005*; *Knovich et al., 2009*). Ferritin in both mammals and insects is a heteropolymer, consisting of heavy and light chain homolog subunits. In *Drosophila*, these subunits are encoded by *ferritin 1 heavy chain homolog* (*Fer1HCH*) and *ferritin 2 light chain homolog* (*Fer2LCH*), respectively (*Arosio et al., 2009*; *Pham and Winzerling, 2010*), which both contain secretion signal peptides that direct them to the ER during translation (*Lind et al., 1998*). Consequently, *Drosophila* ferritin is confined to the secretory pathway and abundant in the hemolymph (*Nichol et al., 2002*). Ferritin or iron manipulation in *Drosophila* glia regulates the adult behavior, such as locomotion (*Kosmidis et al., 2011*; *Navarro et al., 2015*). Previous researches have shown that *Drosophila* ferritin not only directly participates in dietary iron absorption by removing iron from enterocytes across the basolateral membrane, but competes with transferrin to deliver iron between the gut and the fat body (*Tang and Zhou, 2013*;

*Xiao et al., 2019*). These findings underscore the pivotal role of ferritin in iron transport. Interestingly, iron-loaded ferritin, serving as an essential mitogen, promotes proliferation of cultured *Drosophila* cells (*Li, 2010*). Therefore, it is tempting to investigate whether ferritin is required for self-renewal and the proliferation of NSCs during brain development.

Here, we show that glial ferritin supplies iron into NBs to form iron-sulfur (Fe-S) clusters, which is necessary for the production of ATP, providing the energy required for self-renewal and proliferation of NBs, highlighting ferritin as a potential target for suppressing tumor.

## Results

### Glial ferritin is required for NB maintenance and proliferation

To identify new niche factors involved in the balance between NB self-renewal and differentiation, we used *repo-GAL4* to perform an RNAi-mediated screen in glial cells. *Fer1HCH* and *Fer2LCH*, two ferritin subunit encoding genes, were identified to be essential for NB maintenance and brain development. Specifically, knocking down *Fer1HCH* or *Fer2LCH* in glial cells significantly reduced the number of NBs in the CB and VNC at the third-instar larval stage, which resulted in decreased size of the CNS (*Figure 1A and B* and *Figure 1—figure supplement 1A*). To further assess the impact on NB proliferation, we employed the mitotic marker phosphohistone H3 (PH3) and the thymidine analogue 5-ethynyl-2′-deoxyuridine (EdU), revealing a substantial decrease in the proliferation rate of NBs upon glial ferritin knockdown (*Figure 1A and C* and *Figure 1—figure supplement 1B and C*). These results indicated that glial ferritin is required for maintaining NBs and their proliferation.

To identify the glial subpopulation in which ferritin functions, we employed subtype-restricted GAL4 to perform ferritin knockdowns (*Table 1*). We observed that knocking down ferritin genes in cortex glia led to a similar yet milder phenotype compared to that observed in the pan-glia knockdown (*Figure 1—figure supplement 1D*). This milder effect may be attributed to potential compensatory mechanisms from other glial subpopulations. Interestingly, knocking down ferritin in other glial subtypes did not yield any discernible phenotype (*Figure 1—figure supplement 2*). These findings suggest that ferritin functions collectively across all glial populations to regulate the activity of NBs.

We confirmed the phenotype with multiple independent RNAi lines for *Fer1HCH* or *Fer2LCH*, and observed similar abnormalities of larval NBs and brain development (*Figure 1—figure supplement 3A*). Hereafter, *UAS-Fer1HCH-RNAi* (THU5585) and *UAS-Fer2LCH-RNAi* (TH01861.N) were used for most studies, and we only quantified NB number and proliferation rate in the CB for convenience in this research.

To evaluate the efficiency of RNAi for *Fer1HCH* or *Fer2LCH*, RT-PCR was performed with total RNA isolated from the whole brain dissected from third-instar larvae. The abundance of *Fer1HCH* or *Fer2LCH* mRNA was significantly reduced in glial *Fer1HCH* or *Fer2LCH* knockdown, respectively (*Figure 1D and E*). Western blot analysis of larval brain protein extract using polyclonal antibodies against Fer1HCH or Fer2LCH further confirmed that both subunits were down-regulated in glial knockdown of either subunit (*Figure 1F–I*), consistent with previous research in *Drosophila* (*Missirlis et al., 2007*). To exclude the possibility of RNAi off-target effects, we used *repo-GAL4* to overexpress RNAi-resistant *Fer1HCH* (or *Fer2LCH*) containing silent mutations in the region targeted by shRNA, which resulted in a significant increase in the number and proliferation rate of NBs compared to glial ferritin knockdown alone (*Figure 1J–L* and *Figure 1—figure supplement 3B and C*). Additionally, similar but weaker phenotypes were observed when Fer1HCH$^{DN}$, a ferroxidase-inactive form of Fer1HCH (*Missirlis et al., 2007*), was overexpressed in glial cells (*Figure 1—figure supplement 3D*). These results support the conclusion that ferritin in glial cells is essential for the maintenance and proliferation of NBs, suggesting that iron homeostasis is involved in this process.

### Ferritin is produced mainly in glial cells and secreted into NBs through a vesicle-dependent pathway

To investigate the role of iron homeostasis in *Drosophila* CNS, we tried to knock down genes involved in iron transport, storage, and regulation in NBs, neurons, and glial cells, respectively. Interestingly, only glial knockdown of ferritin-related genes led to NB loss and defective brain development. It appears that ferritin generated in glial cells is the only crucial protein for regulating the behavior of NBs and, ultimately, the larval brain development. To recapitulate the endogenous ferritin expression

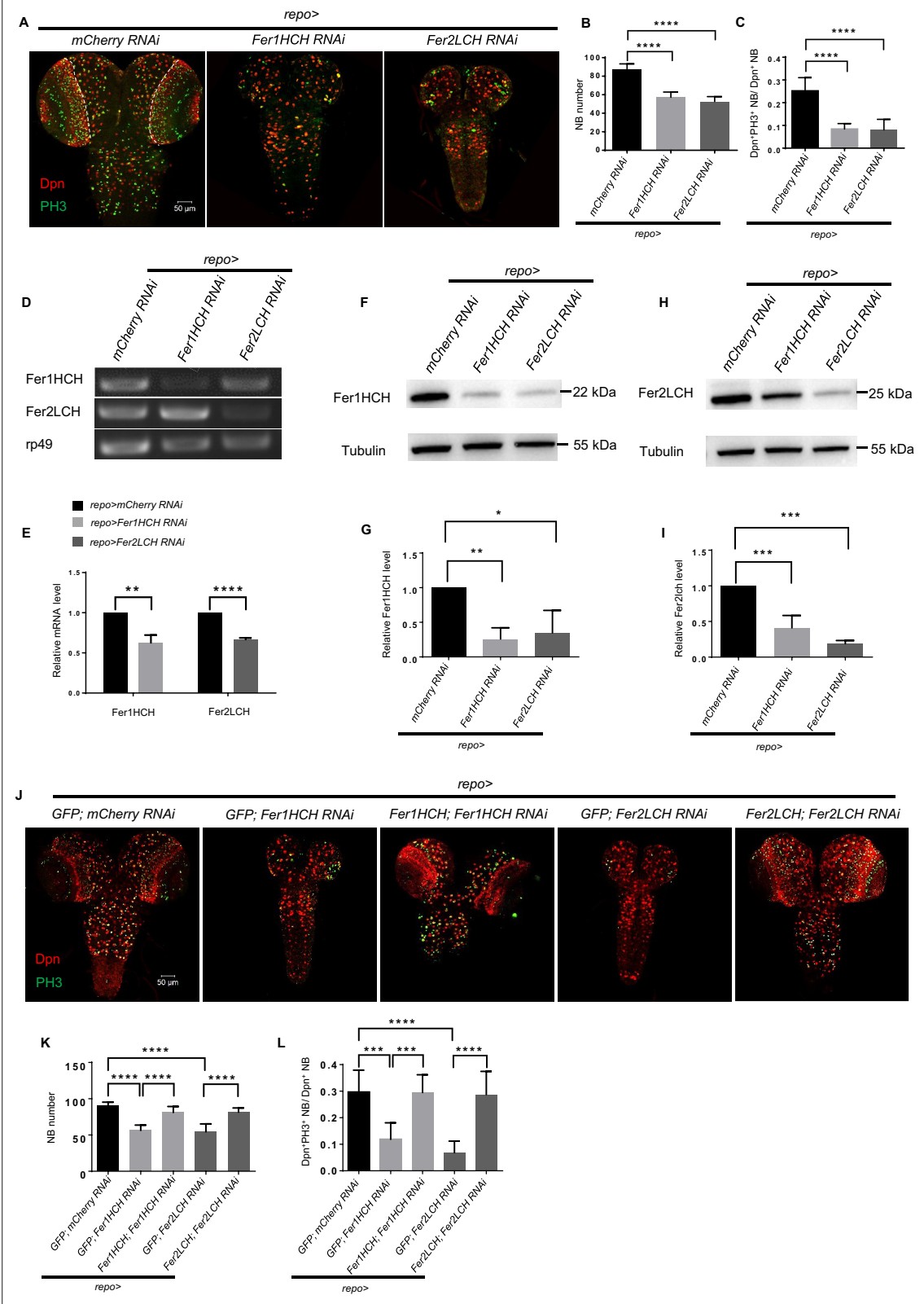

**Figure 1.** Ferritin in the glia is required for neuroblast (NB) maintenance and proliferation. (**A**) NB number and proliferation rate after *Fer1HCH* or *Fer2LCH* RNAi in glia. (**B**) Quantification of NB number in central brain (CB) (**A**). (**C**) Quantification of NB proliferation rate in CB (**A**). (**D and E**) RT-PCR analysis of Fer1HCH and Fer2LCH mRNA levels (**D**) and normalized quantification (**E**) after glial ferritin knockdown. (**F and H**) Western blot of Fer1HCH (**F**) and Fer2LCH (**H**) after ferritin knockdown in glia. Tubulin was used as a loading control. (**G and I**) Quantification of normalized Fer1HCH intensity in

*Figure 1 continued on next page*

*Figure 1 continued*

(**G**) and Fer2LCH intensity in (**I**). (**J**) Rescue of NB number and proliferation when simultaneously overexpressing RNAi-resistant *Fer1HCH* and *Fer1HCH RNAi* (or *Fer2LCH* and *Fer2LCH RNAi*) in glia. (**K** and **L**) Quantification of NB number and proliferation rate in (**J**). (B, n=11; C, n=10; E, G and I, n=3; K, n=8, 8, 9, 7, 9; L, n=9, 8, 8, 7, 12; Statistical results were presented as means ± SD, p values in E were performed by unpaired two-sided Student's t test, other p values were performed by one-way ANOVA with a Bonferroni test; ns, not significant; *p<0.05; **p<0.01; ***p<0.001; ****p<0.0001; Dpn, NB nuclei, red; PH3, green; DAPI, nuclei, blue).

The online version of this article includes the following figure supplement(s) for figure 1:

**Figure supplement 1.** Glial ferritin knockdown leads to low neuroblast (NB) proliferation and number.

**Figure supplement 2.** Ferritin knockdown in different glial subpopulations did not induce any discernible phenotype.

**Figure supplement 3.** Verify the phenotype induced by glial ferritin knockdown using different manipulations.

pattern, we took advantage of a Fer1HCH/Fer2LCH enhancer trapped GAL4 line to drive a variant of DsRed protein with a nuclear localization signal (DsRed Stinger). Surprisingly, our immunostaining experiments revealed that most, if not all, DsRed was co-localized with the glia nuclear protein Repo, instead of the NB nuclear protein Dpn (*Figure 2A and A'*). We also used another Fer2LCH enhancer trapped GAL4 line to verify this result, which was consistent with the above result (*Figure 2—figure supplement 1*). These results further confirm that glial cells are the main producers of ferritin in the larval brain.

*Fer1HCH^G188^* encodes a GFP-tagged Fer1HCH subunit (*Missirlis et al., 2007*), which labels the distribution of Fer1HCH, representing the pattern of ferritin. As shown in the *Figure 2B and B'*, ferritin was detected in NBs besides glial cells, despite that ferritin is specifically produced in glial cells. To explore the possibility of ferritin transfer from glial cells to NBs, we observed the GFP signal of *Fer1HCH^G188^* with glial ferritin knockdown, and found that GFP in the brain was almost completely depleted (*Figure 2B*). Furthermore, we used *repo-GAL4* to drive the expression of GFP-tagged Fer1HCH and mCherry-tagged Fer2LCH in glial cells. We observed that both Fer1HCH::GFP and Fer2LCH::mCherry were present in NBs, while the control GFP and mCherry were restricted to glial cells (*Figure 2C*). Altogether, we conclude that ferritin is generated in glial cells and then transferred to NBs in the larval brain.

In *Drosophila*, both ferritin subunits contain secretion signal peptides that direct protein to the endoplasmic reticulum during translation, which allows the ferritin to be transported out of the cell. To investigate whether the ferritin was transported from glia to NBs via the vesicular transport, we expressed a temperature-sensitive dominant-negative Shibire dynamin protein (shi^ts^) to block vesicle trafficking from glial cells, and found that Fer1HCH::GFP accumulated in glia, accompanying with significantly diminished Fer1HCH signal in NBs (*Figure 2D, D' and E*), suggesting glial ferritin secretes into NBs through vesicle system.

## Glial ferritin defects lead to iron deficiency in NBs

Given that ferritin is required for iron storage as well as delivery, glial ferritin knockdown might increase the free iron level, leading to glial iron overload. Alternatively, it will block the iron transport

**Table 1.** The phenotype of ferritin knockdown in different glial subpopulations.

BDSC: Bloomington *Drosophila* Stock Center; DGRC: *Drosophila* Genetic Resource Center; +++: strong phenotype; ++: weak phenotype; -: no phenotype.

| GAL4 driver | Stock center | Stock number | Expression pattern | Fer1HCH RNAi | Fer2LCH RNAi |
|---|---|---|---|---|---|
| *repo-GAL4* | BDSC | 7415 | all glia | +++ | +++ |
| *nrv2-GAL4* | BDSC | 6799 | cortex glia | + | + |
| *mdr65-GAL4* | BDSC | 50472 | subperineurial glia | - | - |
| *moody-GAL4* | BDSC | 90883 | subperineurial glia | - | - |
| *NP6293-GAL4* | DGRC | 105188 | perineurial glia | - | - |
| *alrm-GAL4* | BDSC | 67032 | astrocyte-like glia | - | - |
| *NP6520-GAL4* | DGRC | 105240 | ensheathing glia | - | - |

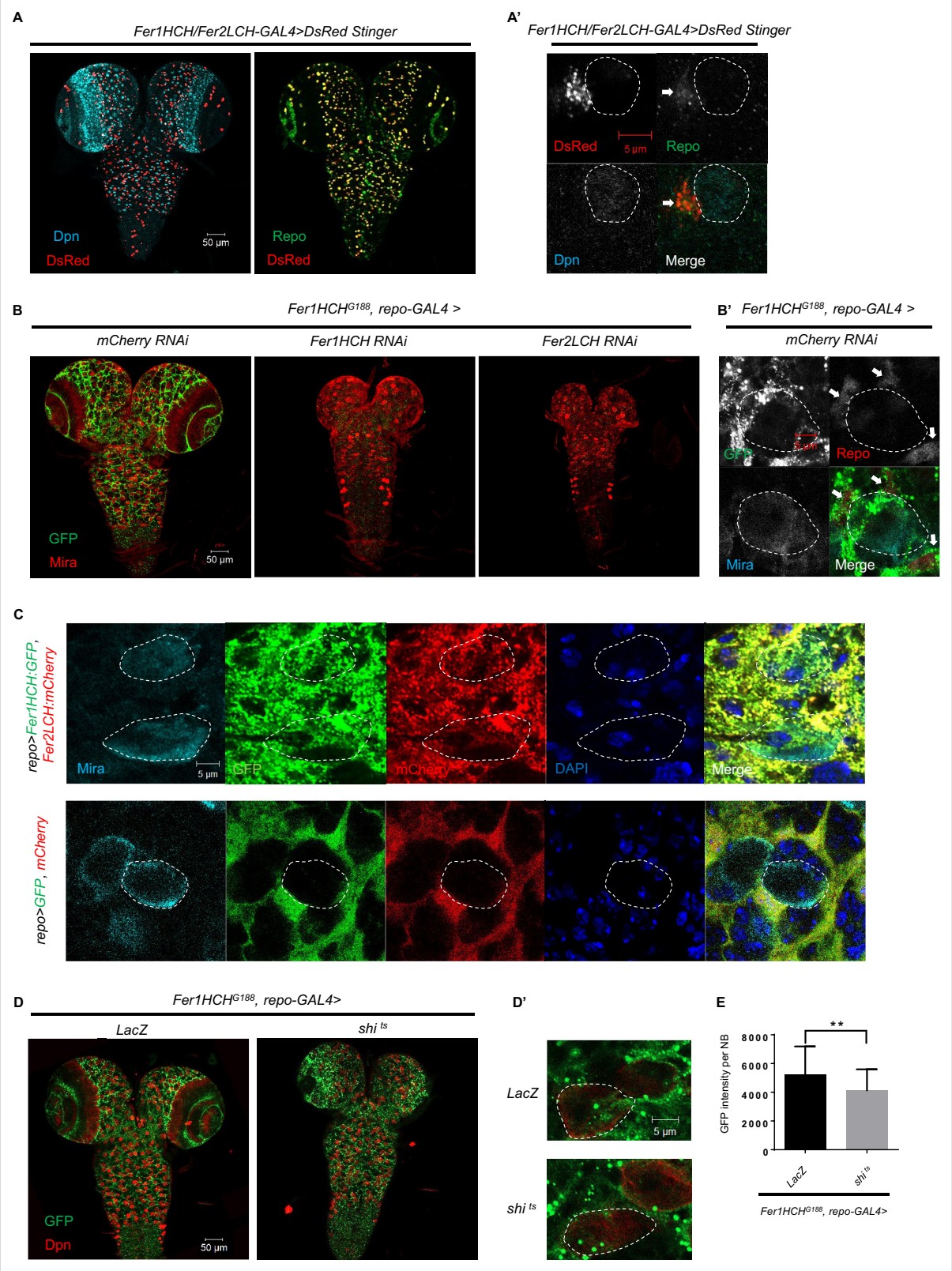

**Figure 2.** Ferritin is produced mainly in glial cells in the *Drosophila* central nervous system (CNS) and secreted into neuroblasts (NBs) through a vesicle-dependent pathway. (**A**) The pattern of DsRed Stinger driven by *Fer1HCH/Fer2LCH-GAL4* in CNS. (**A′**) The magnification of (**A**). (**B**) The distribution of ferritin labeled by *Fer1HCH^G188* in CNS of control and glial ferritin knockdown. (**B′**) The magnification of the left panel in (**B**). (**C**) Overexpressed ferritin tagged with GFP and mCherry in glia secreted into NBs. (**D**) The distribution of *Fer1HCH^G188* when blocking glial vesicular trafficking via overexpressing

*Figure 2 continued on next page*

*Figure 2 continued*

dominant-negative dynamin. (**D′**) The magnification of (**D**). (**E**) Quantification of (**D**), GFP signal in NBs when overexpressing *shi* ᵗˢ in glia. (All white circles indicate NB position and white arrows point to glial cell; E, n=50, 48; Statistical results were presented as means ± SD, p value was performed by unpaired two-sided Student's t test; **p<0.01).

The online version of this article includes the following figure supplement(s) for figure 2:

**Figure supplement 1.** *Fer2LCH-GAL4* was used to validate the pattern of *Fer1HCH/Fer2LCH-GAL4*.

---

from glia to NBs, resulting in NB iron deficiency. To determine the underlying cause of the brain development defects observed upon glial ferritin knockdown, we used iron chelator or iron salt ferric ammonium citrate (FAC) to manipulate iron intake and detect how it will influence NBs. We observed that depriving iron intake by adding iron chelator bathophenanthrolinedisulfonic acid disodium (BPS) to the food led to a dramatic decrease in the number and proliferation rate of NBs, compared to control larvae on normal food (*Figure 3A*). To investigate the effect of feeding behavior in different foods, we tested the feeding behavior of flies by incorporating 1% Brilliant Blue (sigma, 861146) into different foods (*Tanimura et al., 1982*) and found the amount of dye in the fly body was similar between normal group and BPS group (*Figure 3—figure supplement 1A*), which suggested that BPS almost did not affect the feeding behavior. In line with this result, blocking iron absorption by knocking down intestinal ferritin (*Tang and Zhou, 2013*) resulted in the similar defects of the brain (*Figure 3—figure supplement 1B–D*). However, supplementing the diet with FAC did not induce any observable defects (*Figure 3A*). We also tested another iron chelator deferiprone (DFP) (*Soriano et al., 2013*), and found that DFP alone did not cause any brain defects in wild-type (WT) larvae, but it significantly exacerbated the effect of glial ferritin knockdown on defective NB maintenance (*Figure 3B and C*). Furthermore, we attempted to rescue the brain defects observed in glial ferritin knockdown by providing dietary FAC, but the effect was negligible (*Figure 3—figure supplement 1E and F*). This may be due to the possibility that the knockdown of glial ferritin largely, if not completely, prevented the transport of iron from glia to NBs, which hindered the additional iron from entering NBs. These results indicate that the NB loss phenotype induced by glial ferritin knockdown is primarily due to iron deficiency in NBs.

Zip13, generally known as a zinc transporter, functions as an iron transporter, cooperating with ferritin on iron transport in the *Drosophila* intestine (*Xiao et al., 2014*). Specifically, the knockdown of *Zip13* leads to failure of iron loading in secretory vesicles and consequently defective iron efflux (*Xiao et al., 2014*). To investigate whether the NB loss phenotype of glial ferritin knockdown is caused by defective iron transport, we next sought to identify the role of Zip13 in the larval brain. First, we examined the expression pattern of Zip13 and found that, similar to ferritin, the DsRed stinger driven by *Zip13-GAL4* was co-localized with Repo in glial cells rather than Dpn in NBs (*Figure 3D and D′*). Subsequently, we knocked down *Zip13* in glial cells and observed a similar but weaker phenotype with a slightly reduced NB number and proliferation rate (*Figure 3E–G*). Consistently, the defective brain development was confirmed in the *Zip13* mutant (*Figure 3—figure supplement 1G–I*). Moreover, the effect of double knockdown of both *Zip13* and *Fer2LCH* was much more severe than that of either single knockdown, with a significant reduction in NB number and proliferation rate (*Figure 3E–G*). Furthermore, we found that the reduced proliferation rate caused by *Zip13* knockdown could be partially rescued by iron supplementation in the food (*Figure 3H and I*). Overall, these results suggest that glial ferritin collaborates with Zip13 to deliver iron from glia to NBs.

Since previous studies have demonstrated the essential role of glial cells in NB survival and proliferation (*Bailey et al., 2015*; *Read, 2018*), it is possible that iron overload in glial cells up-regulated the level of reactive oxygen species (ROS), leading to apoptosis of glial cells, which ultimately resulted in the loss of NBs. To rule out this possibility, firstly, we examined the ROS level of glial ferritin knockdown via a transgenic reporter GstD-GFP for oxidative stress signaling (*Sykiotis and Bohmann, 2008*). GstD1 is a prototypical oxidative stress response gene (*Sawicki et al., 2003*). We found an obvious increase as expected (*Figure 3—figure supplement 2A and B*). However, overexpressing anti-oxidant genes in glial cells or NBs did not restore the NB number and proliferation (*Figure 3—figure supplement 2C and D*). ROS accumulation leads to lipid peroxidation, which in turn results in ferroptosis (*Hirschhorn and Stockwell, 2019*). Mitochondria in cells undergoing ferroptosis typically exhibit a characteristic shrinkage and damage, while the nuclei remain structurally intact (*Mumbauer et al., 2019*). In order to investigate whether the NB loss is due to ferroptosis, we examined the mitochondrial morphology

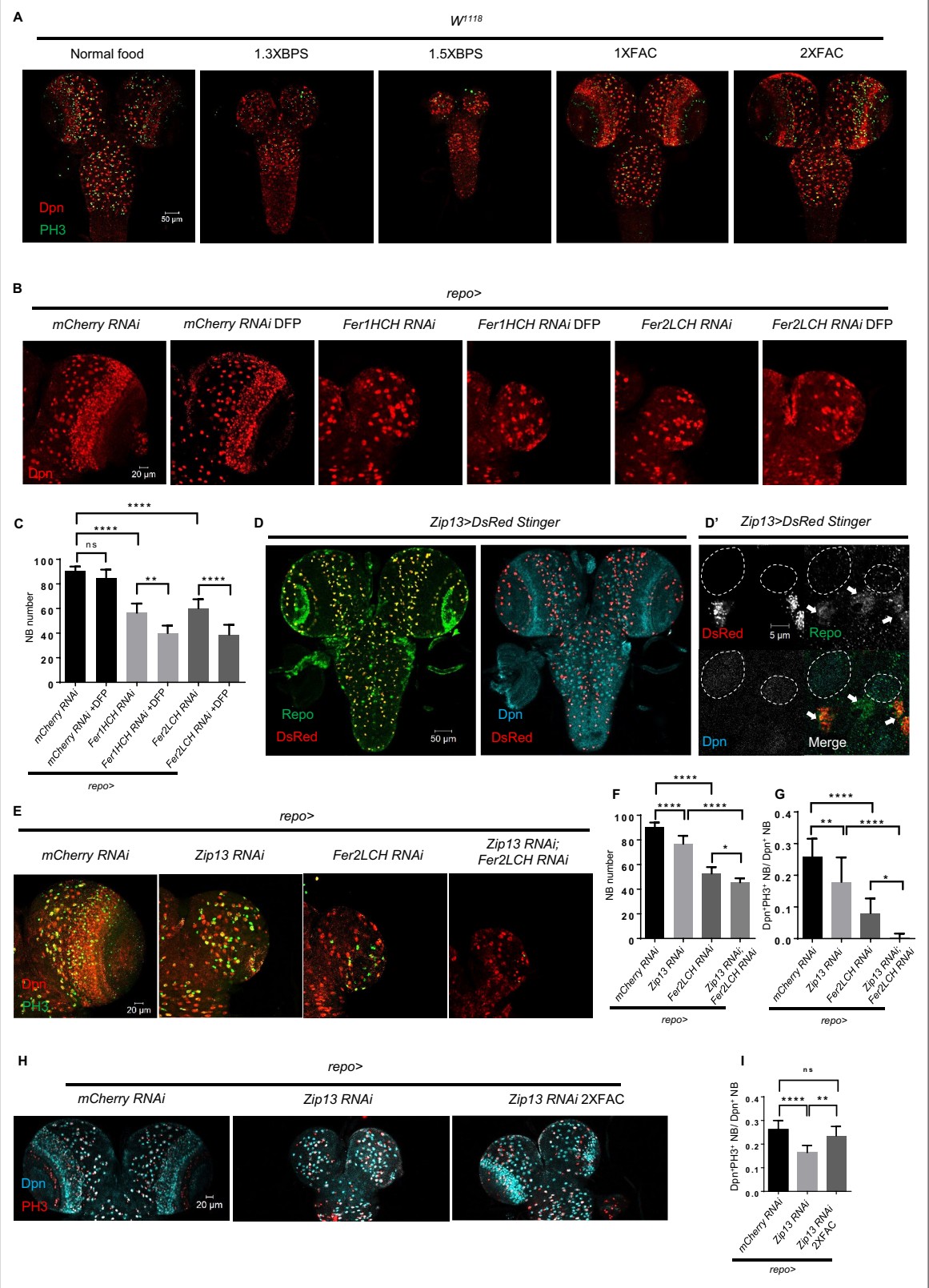

**Figure 3.** Glial ferritin defects lead to iron deficiency in neuroblasts (NBs). (**A**) The conditions of brain development in *Drosophila* with feeding iron chelator bathophenanthrolinedisulfonic acid disodium (BPS) or supplementary iron ferric ammonium citrate (FAC). (**B**) Iron depletion by adding deferiprone (DFP) to food exacerbated the phenotype of NB loss induced by glial ferritin knockdown. (**C**) Quantification of NB number in (**B**). (**D**) Zip13 in central nervous system (CNS) is produced mainly by glial cells, but not NBs. (**D′**) The magnification of (**D**). (All white circles indicate NB position and

*Figure 3 continued on next page*

*Figure 3 continued*

white arrows point to glial cells). (**E**) Brain defects in *Zip13* knockdown or double knockdown of *Zip13* and *Fer2LCH*. (**F and G**) Quantification of NB number and proliferation rate in (**E**). (**H**) Iron supplements can rescue the NB proliferation in *Zip13 RNAi* driven by *repo-GAL4*. (**I**) Quantification of NB proliferation in (**H**). (C, n=6; F, n=6, 10, 11, 8; G, n=13, 10, 10, 8; I, n=8; Statistical results were presented as means ± SD, p values were performed by one-way ANOVA with a Bonferroni test; ns, not significant; *p<0.05; **p<0.01; ****p<0.0001).

The online version of this article includes the following figure supplement(s) for figure 3:

**Figure supplement 1.** Brain defects induced by glial ferritin knockdown were caused by iron deficiency.

**Figure supplement 2.** Reactive oxygen species (ROS) accumulates in central nervous system (CNS) after glial ferritin knockdown, but inhibiting ROS cannot restore neuroblast (NB) number and proliferation.

**Figure supplement 3.** Glial ferritin knockdown does not induce apoptosis in the central nervous system (CNS).

using Transmission Electron Microscopy (TEM). However, our result showed that the mitochondrial double membrane and cristae were clearly visible whether in the control group or glial ferritin knockdown group, which suggested that ferroptosis was not the main cause of NB loss upon glial ferritin knockdown (*Figure 3—figure supplement 2E and F*). In addition, we investigated the level of 4-HNE that indicates lipid peroxidation, the other hallmark of ferroptosis. This result showed that the 4-HNE level did not change significantly after ferritin knockdown (*Figure 3—figure supplement 2G and H*), suggesting that lipid peroxidation was stable, which supported the exclusion of the ferroptosis in glial ferritin knockdown. Taken with the findings from iron chelator feeding together, we conclude that NB loss resulting from glial ferritin knockdown is not a consequence of ferroptosis. Second, we examined the apoptosis in larval brain cells via Caspase-3 or TUNEL staining, and found the apoptotic signal remained unchanged after glial ferritin knockdown (*Figure 3—figure supplement 3A–D*). Furthermore, we overexpressed the baculovirus protein P35 to inhibit apoptosis in glial cells, but failed to restore the lost NBs (*Figure 3—figure supplement 3E*). Given that glial morphology is important for NB lineages (*Spéder and Brand, 2018*), we used GFP driven by *repo-GAL4* to denote glia cells and found that the glial chamber was almost intact after glial ferritin knockdown (*Figure 3—figure supplement 3F*). Together, we conclude that it is iron deficiency in NBs, rather than iron overload in glial cells, that leads to NB loss induced by ferritin defects.

## Glial ferritin defects result in impaired Fe-S cluster activity and ATP production

To further prove that defective iron transport leads to the iron deficiency in NBs, we examined the iron availability by measuring aconitase activity in the brain. The activity of aconitase is sensitive to the level of Fe-S synthesis, so it can serve as an indicator of the availability of iron in the cell (*Haile et al., 1992*; *Tong and Rouault, 2006*). Aconitase activity is determined in a coupled enzyme reaction in which citrate is converted to isocitrate by aconitase (Aconitase Activity Assay Kit, sigma, MAK051). As shown in the *Figure 4A*, aconitase activity of ferritin knockdown diminished significantly, indicating that the availability of iron in the brain was reduced. This result also implies the Fe-S clusters in NBs are probably deficient when knocking down ferritin in glia. Fe-S clusters are essential to activate the aconitase that converts citrate into isocitrate in the tricarboxylic acid cycle (TCA cycle) in mitochondria, and are one type of critical electron carrier for electron transport chain (ETC), especially for the Complex I, II and III function. Considering that the TCA cycle and ETC are critical biological processes for ATP production, we proposed that the knockdown of glial ferritin might lead to ATP deficiency due to the block of the TCA cycle and ETC, which resulted in the failure of maintaining NBs and their proliferation. To verify this hypothesis, we first assessed the function of Fe-S clusters in NBs by knocking down mitochondrial Fe-S protein Nfs1 and the components of the cytosolic iron-sulfur protein assembly (CIA) complex. After knocking down *Nfs1* and the CIA component *CG17683*, *galla2*, and *ciao1* in NBs respectively, the proliferation level of NBs decreased significantly and brain size became smaller (*Figure 4B and C*), which phenocopied the knockdown of *Fer1HCH* or *Fer2LCH* in glia. This result implies that Fe-S clusters are indispensable for NBs to maintain their proliferation.

To identify the biological processes related to Fe-S clusters in NB lineages, we performed transcriptome analysis of sorted NB lineages from glial ferritin knockdown as well as the control groups. We used NB driver *Insc-GAL4* to express *UAS-DsRed*, labeling NB lineages, and then sorted these cells via fluorescence-activated cell sorting (FACS) (*Figure 4D*; *Harzer et al., 2013*). We found that thousands

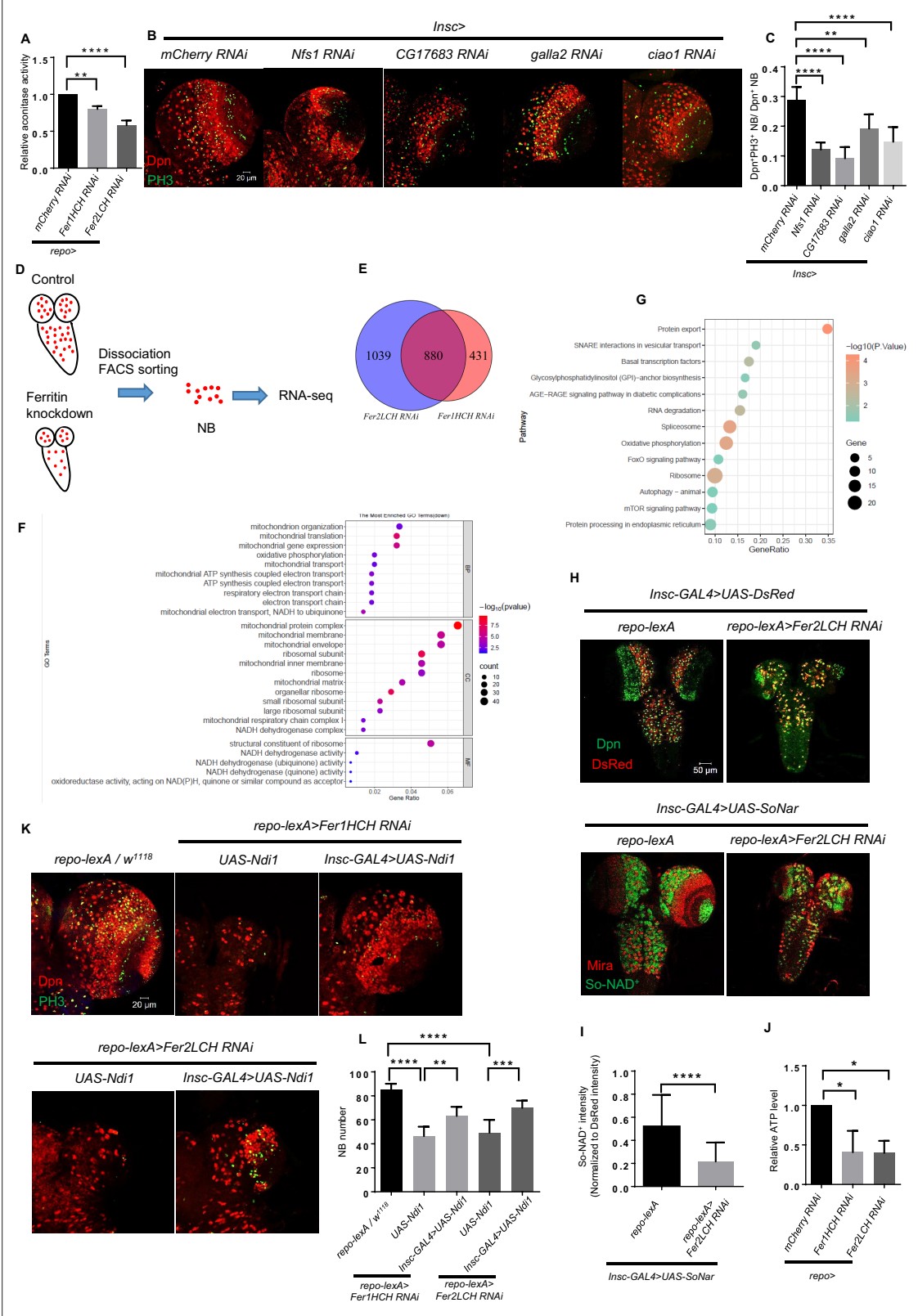

**Figure 4.** Glial ferritin defects result in impaired iron-sulfur (Fe-S) cluster activity and ATP production. (**A**) The determination of cytosolic aconitase activity. (**B**) Knockdown of mitochondrial iron-sulfur protein Nfs1 and cytosolic iron-sulfur protein assembly (CIA) complex in neuroblasts (NBs). (**C**) Quantification of NB proliferation in (**B**). (**D**) The schematic diagram of the NB sorting procedure. (**E**) Venn diagram of downregulated genes after glial ferritin knockdown. (**F**) Gene Ontolog (GO) enrichment of down-regulated genes in glial ferritin knockdown compared with control. (**G**) Kyoto

*Figure 4 continued on next page*

*Figure 4 continued*

Encyclopedia of Genes and Genomes (KEGG) pathway enrichment of down-regulated genes after glial ferritin knockdown. (**H**) NAD$^+$ level was indicated by SoNar. (**I**) Quantification of normalized SoNar signal in (**H**). (**J**) The determination of ATP level in central nervous system (CNS). (**K**) NB number in glial ferritin knockdown was rescued by Ndi1 overexpression in NBs. (**L**) Quantification of NB number in (**K**). (A and J, n=3; C, n=5, 7, 6, 6, 9; I, n=37,42; L, n=7, 7, 6, 6, 7; Statistical results were presented as means ± SD, p value in I was performed by unpaired two-sided Student's t test, other p values were performed by one-way ANOVA with a Bonferroni test; *p<0.05; **p<0.01; ***p<0.001; ****p<0.0001).

The online version of this article includes the following figure supplement(s) for figure 4:

**Figure supplement 1.** Validation of RNA-seq data by qRT-PCR using sorted neuroblast (NB) lineages and Ndi1 overexpression.

of genes were significantly altered in the *Fer1HCH* and *Fer2LCH* knockdown groups compared to the control group. Specifically, 880 genes were simultaneously downregulated among the significantly changed genes in both the *Fer1HCH* and *Fer2LCH* knockdown groups contrasted to the control group (*Figure 4E*). Given that iron deficiency in NBs probably leads to the defects of energy production and biosynthesis, we performed gene enrichment analysis in these genes based on Gene Ontology (GO) and the Kyoto Encyclopedia of Genes and Genomes (KEGG). GO term analysis showed that mitochondrial organization, translation, transport, oxidative phosphorylation (OxPhos), and ETC were enriched in the Biological Processes (*Figure 4F*), indicating NB mitochondrial functions including OxPhos and ETC were abnormal after glial ferritin knockdown. Moreover, NADH dehydrogenase and ribosome-related terms were also enriched whether in the Cellular Component or Molecular Function (*Figure 4F*), suggesting that NADH dehydrogenase and biosynthesis were defective. Further, we validated the enriched term associated with mitochondrial function by qRT-PCR (*Figure 4—figure supplement 1A*). And the enriched KEGG pathways included OxPhos and ribosome (*Figure 4G*), aligning with the GO enrichment results and supporting the hypothesis that glial ferritin knockdown disrupted mitochondrial function including OxPhos and ETC as well as further biosynthesis.

To confirm whether the function of NADH dehydrogenase was disrupted, we used the SoNar that could bind to NAD$^+$ with a specific conformation and excitatory wavelength to monitor the NAD$^+$ level (*Bonnay et al., 2020*; *Zhao et al., 2016*). SoNar was driven by the *Insc-GAL4* to indicate the NAD$^+$ level in NBs. Considering the possibility of protein accumulation in NBs due to the lower proliferation rate in the glial ferritin knockdown group, the signal of SoNar was normalized to DsRed Stinger driven by *Insc-GAL4*. We observed significantly declined SoNar/DsRed after knocking down *Fer2LCH* (*Figure 4H and I*), suggesting the normalized level of NAD$^+$ in NBs was decreased, which supports the conclusion that the activity of NADH dehydrogenase was declined. To further verify the change in energy production, we utilized the ATP Determination Kit which offers a convenient bioluminescence assay for the quantitative determination of ATP with recombinant firefly luciferase and its substrate D-luciferin. We found that ATP level decreased significantly after glial ferritin knockdown when compared to the control (*Figure 4J*). Therefore, we tried to supply the energy for NBs by restoring bioenergy-dependent NAD$^+$ metabolism. To validate this, we first determined the ATP production after overexpressing Ndi1 in NBs upon glial *Fer2LCH* knockdown. The data suggested that expression of Ndi1 can restore ATP production (*Figure 4—figure supplement 1B*). Furthermore, the NB number could be partially rescued by overexpressing proton-pump-independent yeast mitochondrial NADH dehydrogenase Ndi1 in NBs under the background of glial ferritin knockdown (*Figure 4K and L*; *Bonnay et al., 2020*). In summary, the iron deficiency leads to NB loss due to the insufficient energy required for essential biological processes.

## Glial ferritin maintains NBs by preventing premature differentiation

To investigate how glial ferritin knockdown leads to NB loss, we first examined whether NB origin was affected by altering the timing of ferritin knockdown. We used the temperature-sensitive GAL4 inhibitor GAL80$^{ts}$ to suppress ferritin RNAi at a permissive temperature (18°C) during the embryonic stage, and then shifted to a restrictive temperature (29°C), or vice versa until we dissected them at the third-instar larval stage. Ferritin knockdown during the larval stage led to severe brain development defects (*Figure 5A and B*), while the knockdown only during the embryonic stage showed no apparent abnormality (*Figure 5—figure supplement 1A and B*). Together, these results indicate that glial ferritin is required for NB maintenance at the larval stage. Consistent with this result, we found no obvious defects of the brain at the first-instar larval stage when knocking down glial ferritin (*Figure 5—figure supplement 1C and D*). Since newborn NBs are delaminated from the neuroectoderm during the

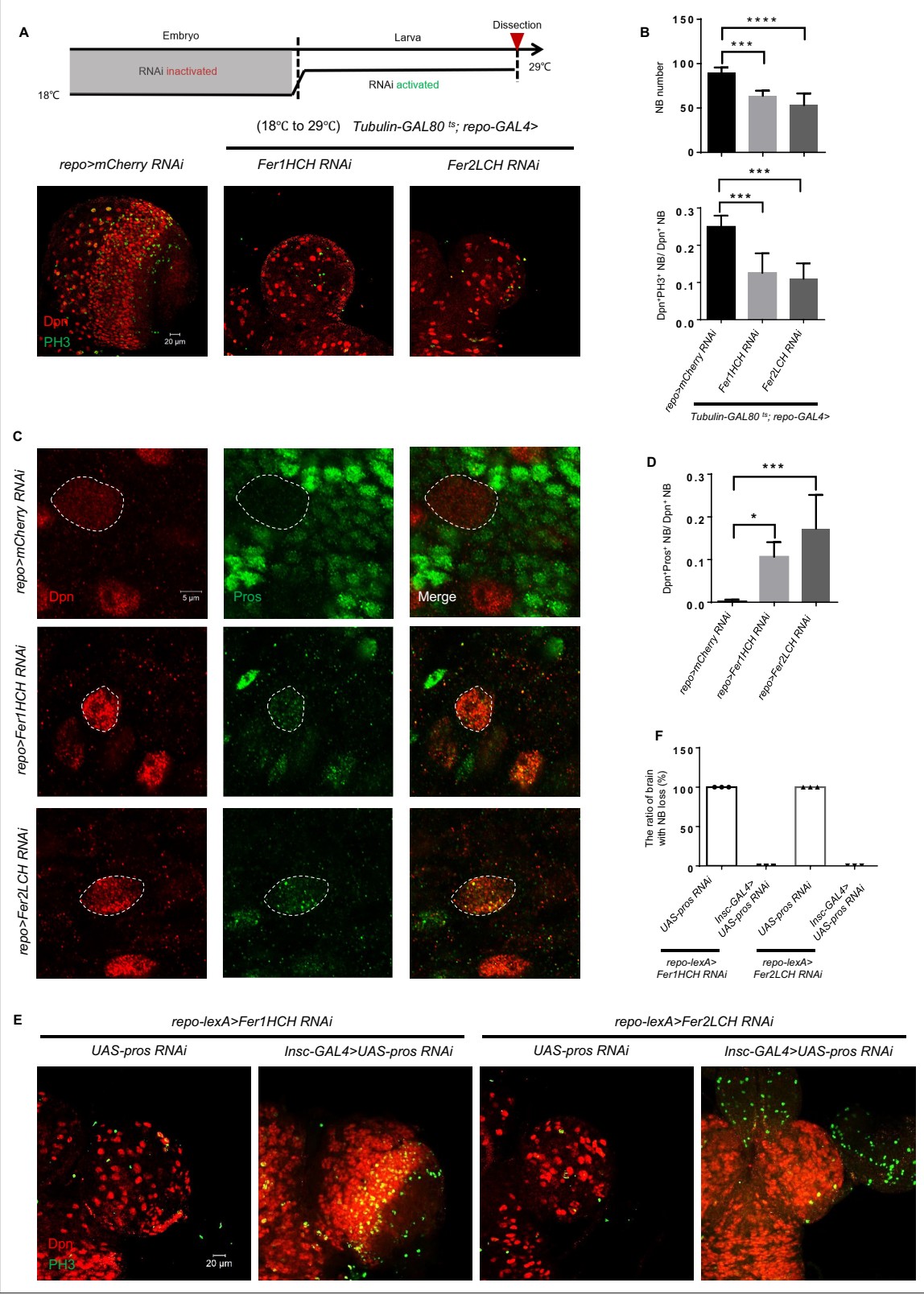

**Figure 5.** Glial ferritin defects lead to the premature differentiation of neuroblasts (NBs). (**A**) Temporal window of glial ferritin maintaining NBs. The temperature shift of flies with glial ferritin knockdown from the permissive temperature (18℃) to the restrictive temperature (29℃). (**B**) Quantification of NB number and proliferation in (**A**). (**C**) Pros staining of knocking down glial ferritin. (**D**) Quantification of the ratio of Pros⁺Dpn⁺ NBs to Dpn⁺ NBs.

*Figure 5 continued on next page*

*Figure 5 continued*

(**E**) Knockdown of *pros* in NBs rescues the NB number. (**F**) The ratio of the brain with NB loss in (**E**). (B, n=6, 5, 6; n=6, 5, 4; D, n=5; F, n=3; Statistical results were presented as means ± SD, p values were performed by one-way ANOVA with a Bonferroni test;*p<0.05; ***p<0.001; ****p<0.0001).

The online version of this article includes the following figure supplement(s) for figure 5:

**Figure supplement 1.** Neuroblast (NB) loss upon glial ferritin knockdown is not due to NB origin and apoptosis.

**Figure supplement 2.** Ndi1 overexpression can restore Pros localization.

**Figure supplement 3.** Glial ferritin defects do not affect neuroblast (NB) reactivation and asymmetric division.

embryonic stage (stage 9–11), and ferritin RNAi only during the embryonic stage did not show any phenotype, suggesting that NB loss after glial ferritin knockdown is not ascribed to defective NB origin.

Furthermore, we explored the earliest stage when the phenotype (NB number and proliferation) existed after glial ferritin knockdown. The result showed that NB proliferation decreased significantly, but NB number declined slightly at the second-instar larval stage (*Figure 5—figure supplement 1E and F*), suggesting that brain defect of glial ferritin knockdown manifests at the second-instar larval stage.

As mentioned before, we did not detect any apoptotic signal in the larval brain after glial ferritin knockdown, suggesting that there is no association between apoptosis and NB loss. The double binary systems, lexA/lexAop, and GAL4/UAS, were employed to further exclude the possibility of apoptosis. We overexpressed P35 in NBs by *Insc-GAL4* and knocked down ferritin by *repo-lexA*, which failed to restore the decreased number of NBs (*Figure 5—figure supplement 1G*). These results reinforce the notion that NB loss occurs independently of the apoptotic pathway.

Next, we hypothesized that the NB loss might be attributed to nuclear-Prospero (Pros)-dependent premature differentiation of NBs (*Cabernard and Doe, 2009*; *Choksi et al., 2006*). Pros in NB nuclear was hardly detected in third-instar WT larvae, but was increased significantly with glial ferritin knockdown (*Figure 5C and D*). To further verify that the NB loss caused by glial ferritin knockdown was due to nuclear-Pros-dependent premature differentiation, we used the *Insc-GAL4* to drive *UAS-pros-RNAi* in order to prevent the NB loss in glial ferritin knockdown background. We found that the NB number was substantially restored (*Figure 5E and F*). Interestingly, despite of dramatic NB number increase, the decreased proliferation rate was not rescued. We proposed that *pros* knockdown in NBs blocked premature terminal differentiation, but it could not replenish the reduced energy caused by glial ferritin knockdown. As NB self-renewal and proliferation are both energy-consuming processes, the self-renewal was rescued via inhibiting differentiation by *Insc-GAL4* driving *UAS-pros-RNAi*, but at the expense of energy consumption for proliferation, leaving the proliferation rate unchanged, or even lower. Together, we conclude that glial ferritin maintains the number of NBs by preventing the nuclear-Pros-dependent premature differentiation.

Furthermore, we also investigated whether overexpression of Ndi1 could restore Pros localization in NBs. This result showed that overexpressing Ndi1 could significantly restore Pros localization in NBs (*Figure 5—figure supplement 2*), which supported that energy shortage induced the entry of Pros into nuclei, leading to the premature differentiation of NBs.

Considering that asymmetric division defects in NBs may lead to premature differentiation, we also explored the asymmetric division by staining the classical asymmetric marker aPKC and found it displayed a crescent at the apical cortex based on the daughter cell position whether in control or glial ferritin knockdown (*Figure 5—figure supplement 3A*). This result indicated that there was no obvious asymmetric defect after glial ferritin knockdown.

Given that reduced proliferation is a distinctive characteristic of quiescent NBs, we tried to determine whether NBs were in a quiescent state after glial ferritin knockdown by examining the other two key features associated with quiescent NBs: decreased cell size and the formation of the extended cellular protrusions (*Chell and Brand, 2010*; *Ly et al., 2019*). However, we found that the NB size remained at the same level as the control, and the extended cellular protrusion was not observed after glial ferritin knockdown (*Figure 2B* and *Figure 5—figure supplement 3B*). Furthermore, since insulin signaling is necessary and sufficient to reactivate NBs from quiescence (*Sousa-Nunes et al., 2011*), we tried to overexpress a constitutively active form of InR (InR^act) in NBs to promote reactivation at the background of glial ferritin knockdown. However, InR^act overexpression failed to ameliorate the

NB abnormalities including the decreased proliferation (*Figure 5—figure supplement 3C and D*). In conclusion, NBs exhibited a low proliferation rate after glial ferritin knockdown, but were not in a quiescent state.

## Ferritin functions as a potential target for tumor suppression

We have demonstrated that glial ferritin is essential for NB self-renewal and proliferation by providing glia-to-NB iron transport to support energy production in NBs. We next tried to determine whether manipulating this biological function could suppress tumor development. We induced tumor development by driving *UAS-brat-RNAi* or *UAS-numb-RNAi* under the control of *Insc-GAL4*, and knocked down ferritin genes under the control of *repo-lexA*. We observed that the size of tumors was strikingly decreased in the larval brains lacking glial ferritin (*Figure 6A and B* and *Figure 6—figure supplement 1A and B*). This result suggests that glial ferritin is required for tumor development, highlighting a potential target for tumor treatment.

To verify whether iron is involved in the tumor suppression induced by blocking glial ferritin, we tried to decrease iron levels by adding the iron chelator BPS in the *Drosophila* food. Tumor formation in the brain at the third instar larval stage was induced with the same method as mentioned above. The addition of BPS into dietary food significantly reduced the tumor size (*Figure 6C and D* and *Figure 6—figure supplement 1C and D*), suggesting that iron is required for tumor development in *Drosophila* larval brain. Considering the conservation of iron's fundamental function as an essential nutrient for cell growth and proliferation, we thus tried to suppress brain tumors in mice via iron deficiency. Initially, we constructed an orthotopic glioma model in C57BL/6 J mice. One week after inoculation of glioma cells (GL261-luc), the iron chelator DFP was intraperitoneally injected every two days based on the murine weight until being dissected on days 7, 14, and 21, respectively. Histological examination with HE staining revealed that the size of glioma in mice, injected with DFP, decreased significantly on days 14 and 21 (*Figure 6—figure supplement 1E–G*). Since glioma cells GL261-luc carry the luciferase gene, we injected the substrate luciferin potassium salt into the tumor-bearing mice and detected the bioluminescence signal of mouse gliomas using IVIS Spectrum. This result showed that the bioluminescence signal of glioma in mice with the injection of 10 mg/ml DFP was significantly reduced (*Figure 6E and F*). The above results suggested that iron deficiency induced by DFP could suppress glioma in mice effectively. Furthermore, we assessed the survival time of the tumor-bearing mice with DFP or saline injection and found that 10 mg/ml DFP injection could significantly prolong the survival time of tumor-bearing mice (*Figure 6G*). Altogether, blocking iron in the tumor is a potential therapy for tumor suppression.

## Glial ferritin level is regulated by NBs

The aforementioned results have highlighted the significant role of glial ferritin in controlling NB number and proliferation. We next wondered whether NBs could provide feedback to regulate glial ferritin levels. To answer this question, we analyzed ferritin levels after altering NB proliferation. Previous studies have corroborated that NB proliferation was diminished by blocking the Tor signaling pathway or enhanced by knocking down tumor suppressor genes, such as *brat* or *numb* (*Bowman et al., 2008*; *Reichardt et al., 2018*; *Sousa-Nunes et al., 2011*). When NBs were in low proliferative status, ferritin declined slightly with no significance (*Figure 7A–D*). Conversely, highly proliferative NBs, able to form tumors, induced significantly increased levels of the ferritin (*Figure 7A–D*). In summary, the production of ferritin in glial cells depends on the level of NB proliferation.

## Discussion

Here, we demonstrate that glia regulates iron metabolism in the brain to sustain NBs and their proliferation. Glial ferritin coordinates Zip13 to provide iron for NBs. Once ferritin in glia is defective, NBs proliferate much slower and differentiate prematurely, implying that iron is a necessary metallic element for NBs. Iron participates in the synthesis of Fe-S clusters which activate aconitase required for TCA and facilitate electron-transfer processes involved in OxPhos. Since TCA and OxPhos are essential steps of ATP production, glial ferritin supplies iron into NBs to ensure energy production for maintaining NBs and their proliferation (*Figure 7E*).

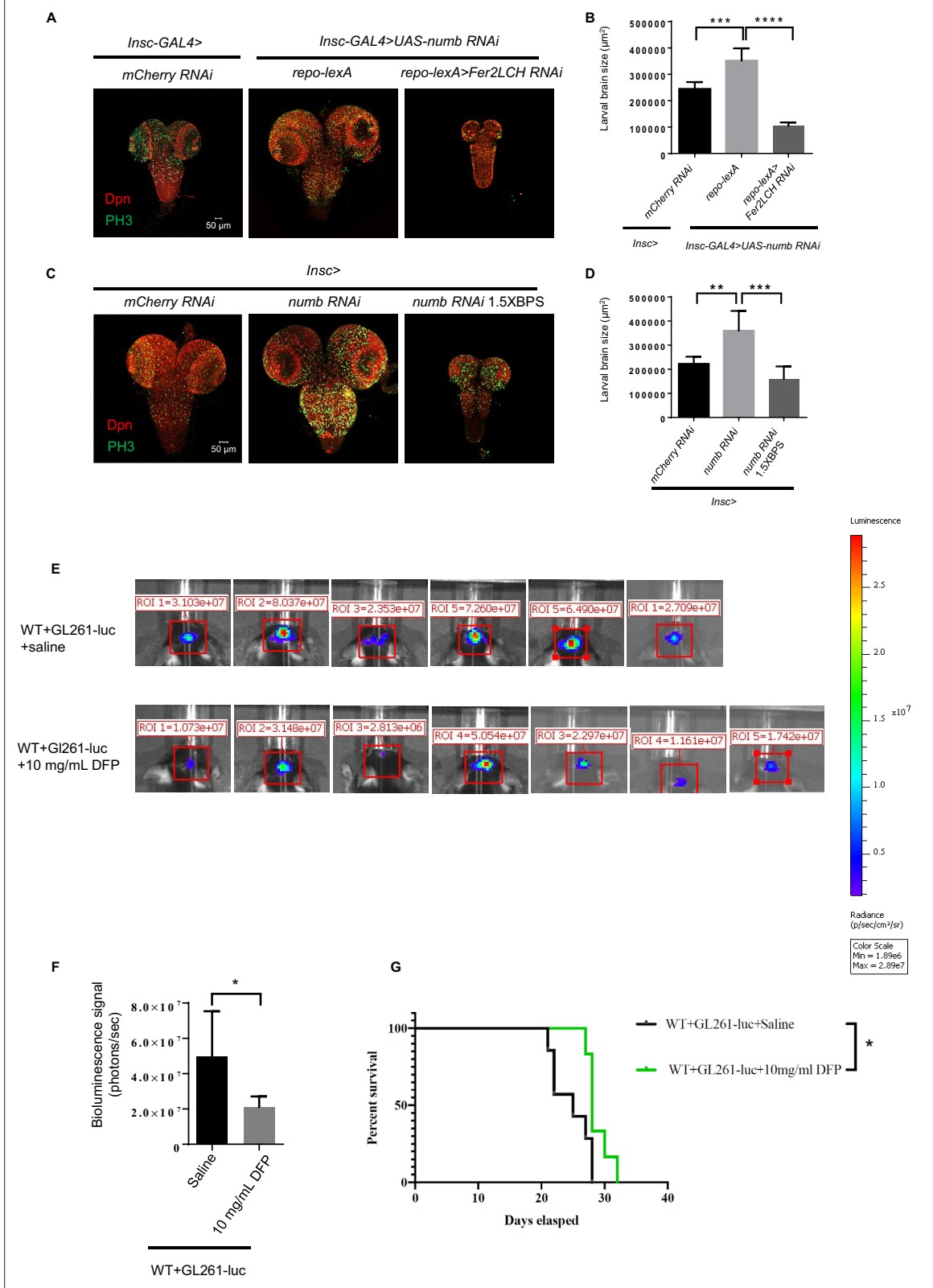

**Figure 6.** Ferritin functions as a potential target for tumor suppression. (**A**) Knockdown of ferritin in glia could inhibit the tumor induced by *numb RNAi*. (**B**) Quantification of larval brain size in (**A**). (**C**) Iron chelator bathophenanthrolinedisulfonic acid disodium (BPS) supplemented in the food suppressed brain tumor growth. (**D**) Quantification of larval brain size in (**C**). (**E**) Bioluminescence signal in mice with glioma on day 14. (**F**) Quantification of bioluminescence signal in (**E**). (**G**) Kaplan-Meier survival curve of mice treated with deferiprone (DFP). (B, n=6, 7, 5; D, n=5, 6, 6; F, n=6, 7; G, n=7,

*Figure 6 continued on next page*

*Figure 6 continued*

6; Statistical results were presented as means ± SD, p values in B and D were performed by one-way ANOVA with a Bonferroni test, p value in F was performed by unpaired two-sided Student's t test, p value in G was performed by log-rank test; *p<0.05; **p<0.01 ***p<0.001; ****p<0.0001).

The online version of this article includes the following figure supplement(s) for figure 6:

**Figure supplement 1.** Iron chelator inhibits tumor progression in *Drosophila* and mice.

## NSCs and glial niche

Previous literatures have reported how the glial niche regulates NB reactivation, proliferation, and survival. Blood-brain barrier (BBB) glia secretes dILP2 and dILP6 that bind on the InR of NBs, which switches on Insulin signaling to reactivate NBs (*Chell and Brand, 2010*; *Spéder and Brand, 2014*). In addition, Pvr (PDGFR ortholog) signaling in cortex glia sustains NB survival and proliferation in a PI3K-DE-cadherin-dependent manner (*Read, 2018*). However, it is unknown whether the glial niche is required for maintaining the self-renewal of NBs. In this study, we demonstrate that the glial niche maintains the self-renewal and proliferation of NBs through supplying the necessary microelement iron into NBs for enough energy production, which implies that mammalian glia likewise provide critical microelement for maintaining NSC function. Oligodendrocytes have been reported to secret ferritin to protect neurons (*Mukherjee et al., 2020*), however, it remains unknown whether the ferritin secretion is employed by oligodendrocytes to regulate NSCs. Astrocytes regulate NSC proliferation via secreting growth factors (*Shetty et al., 2005*), but it is unclear whether ferritin from astrocytes regulates NSC proliferation and other NSC behaviors, such as self-renewal and differentiation. Based on our findings, it is feasible to regulate the self-renewal and differentiation of NSCs indirectly through niche cells, which would help to reduce the risk of directly manipulating NSCs to treat neural diseases.

## Ferritin and iron

Our results show that ferritin generated outside the brain was not detected in the CNS and vice versa (data not shown), suggesting ferritin can hardly cross the BBB glia, possibly because ferritin is too large to permeate. In this way, BBB glia separates the labile systemic iron from local iron inside the brain to guarantee the normal brain development, as the neural cells, including NBs, require a stable iron level for continuous energy generation during neurogenesis.

Although the dominant function of ferritin is iron storage in mammals, increasing evidences suggest that ferritin also can transport iron. H-ferritin expression in astrocytes is necessary for proper oligodendrocyte development and myelination (*Cheli et al., 2021*), which implies H-ferritin probably transports iron from astrocytes to oligodendrocytes for myelination. In mice, ferritin heavy chain is secreted by oligodendrocytes via extracellular vesicles (*Mukherjee et al., 2020*), which possibly endows ferritin with the function of iron transport. Consistently, previous results showed that ferritin can deliver iron across the cultured BBB (*Chiou et al., 2019*; *Fisher et al., 2007*). Our results emphasize the critical role of ferritin in iron transport in the CNS. Future studies would be interesting to investigate whether ferritin is able to transport iron in the mammalian brain.

Ferritin specifically targets on the tumor cells, which has been applied to tumor visualization and targeted drug delivery (*Fan et al., 2012*; *Fan et al., 2018*). Our study shows that the block of ferritin or iron can inhibit tumors effectively, which offers an explanation that ferritin targets on the tumor is to transport iron for maintaining the cell proliferation, highlighting a potential target for tumor suppression. In consideration of the role of ferritin in normal tissue, such as antioxidant defense function for neurons (*Mukherjee et al., 2020*), the entire block of ferritin for suppressing tumor probably brings neurological disorders. Therefore, we should target on ferritin specifically in tumors to suppress tumor growth.

Iron is an essential component for Fe-S clusters which participate in TCA and OxPhos, providing energy required for self-renewal and proliferation of NBs. Previous studies claimed that OxPhos is required for NB proliferation (*van den Ameele and Brand, 2019*). However, another research observed that OxPhos is dispensable in type II NBs (*Bonnay et al., 2020*). Our data imply that energy deficiency in NBs caused by iron shortage leads to NB loss and low proliferation possibly through simultaneous disruption of TCA and OxPhos. Consistently, iron-loaded ferritin nanoparticles can be used to improve the self-renewal ability and differentiation potential of human NSCs in vitro (*Lee et al., 2018*). Additionally, previous studies have shown that iron-loaded ferritin can promote the proliferation of

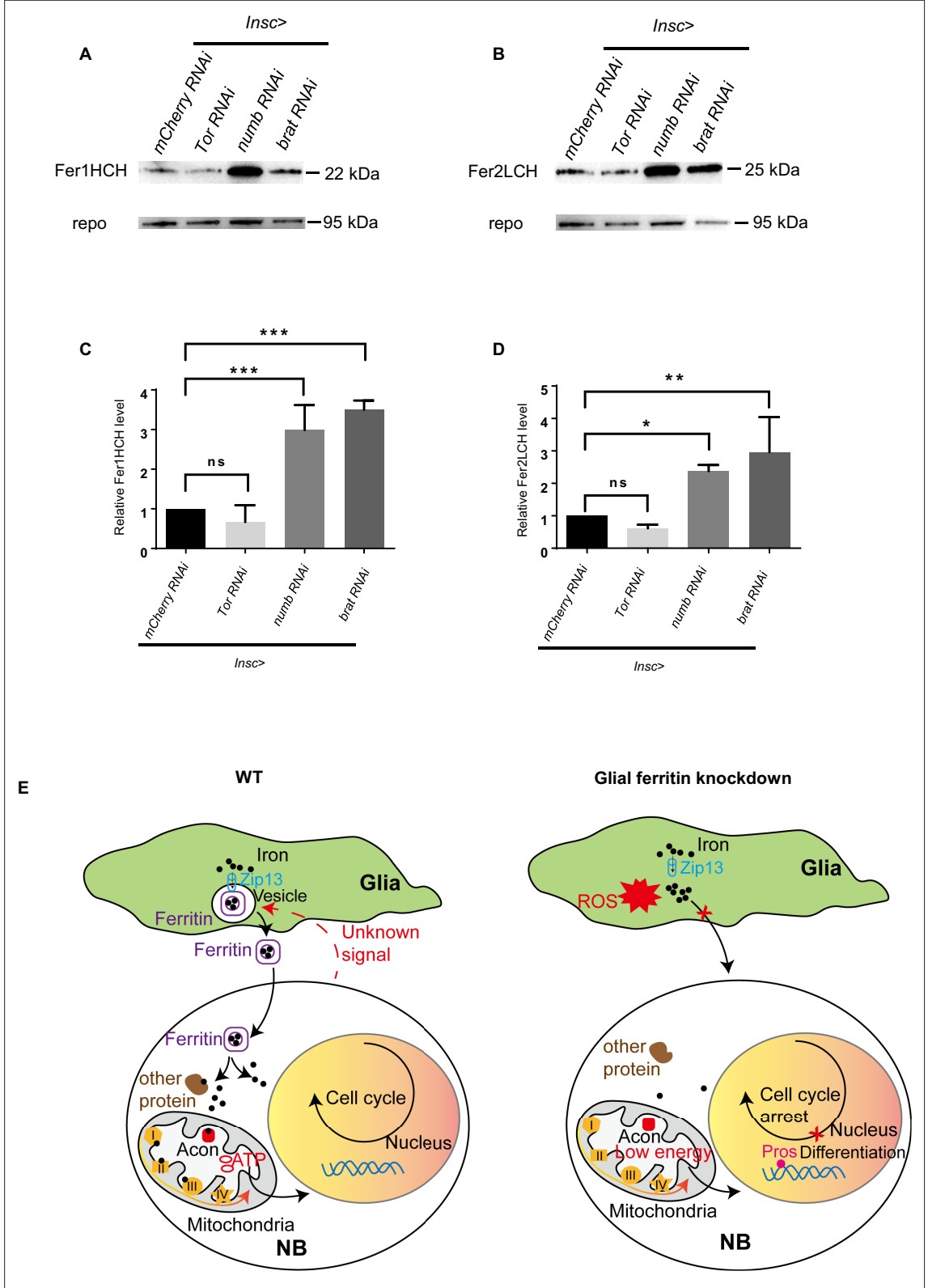

**Figure 7.** The level of ferritin is controlled by neuroblast (NB) proliferation. (**A**) Fer1HCH subunit level in *Drosophila* central nervous system (CNS) after manipulating proliferation. (**B**) Fer2LCH subunit level in *Drosophila* CNS after manipulating proliferation. (**C**) Quantification of Fer1HCH level in (**A**). (**D**) Quantification of Fer2LCH level in (**B**). (**E**) The model of glial ferritin regulating NBs. (C and D, n=3; Statistical results were presented as means ± SD, p values were performed by one-way ANOVA with a Bonferroni test; ns, not significant; *p<0.05; **p<0.01; ***p<0.001).

cultured cell from *Drosophila* (*Li, 2010*). Furthermore, iron depletion was reported to suppress human cancer cells growth at G1/S by inducing cyclin D1 proteolysis (*Nurtjahja-Tjendraputra et al., 2007*). These studies suggest that iron, as a necessary microelement for energy production, is essential to NSC maintenance and proliferation.

## Bicellular iron homeostasis

Iron homeostasis can be regulated systemically by the liver-derived hormone hepcidin and the iron exporter ferroportin, and it is also regulated cellularly by the IRP/IRE system (*Hentze et al., 2010*). Here, we propose a bicellular iron homeostasis between NB and glial niche cell, which is distinct from classic cellular and systemic iron homeostasis. The bicellular iron homeostasis requires an elaborate mechanism to guarantee normal neurogenesis.

In a proliferative state, NBs receive iron from glial cells via ferritin to fulfill the energy requirements for proliferation and self-renewal. Therefore, the iron required by NBs is mainly stored in the ferritin of glial cells and allocated as needed. The bicellular iron homeostasis mechanism makes glial cells the main pool for ferritin production, relieving NBs from the need to produce ferritin and transferring the potential risk of iron overload-induced oxidative stress to glial cells. Furthermore, glial cells can subsequently clear oxidative stress through lipid droplets (*Bailey et al., 2015*), ensuring normal NB function during neural development. It is intriguing to explore whether bicellular iron homeostasis exists in other stem cells and their niches.

When normal NBs are converted into immortal tumor NBs, becoming highly proliferative, they need more nutrients including iron (*Bonnay et al., 2020*; *van den Ameele and Brand, 2019*). In response, more ferritin is generated to transport iron from the glial cells to NBs. However, it is unclear how NBs communicate with glial cells to determine the level of ferritin production. It's interesting to identify the specific cues from NBs that regulate the level of ferritin in future studies. In addition, iron distribution in the *Drosophila* brain is not observed due to low iron concentration and technical limitations in iron detection, which results in a lack of direct evidence for iron trafficking via ferritin.

# Materials and methods
## Fly genetics

Fly strains were bred on yeast-containing medium at a constant of temperature 25°C in 12 hr light/dark cycle unless otherwise stated. UAS-Fer1HCH-RNAi (THU5585) and UAS-Fer2LCH-RNAi (TH01861.N) were used for most experiments except *Figure 1—figure supplement 2A* (BDSC: 60000, VDRC: 14491, and VDRC: 12925) and *Figure 3—figure supplement 3E* (VDRC: 14491). *UAS-Fer1HCH*, *UAS-Fer2LCH*, *lexAop-Fer1HCH-RNAi* and *lexAop-Fer2LCH-RNAi* were generated in this paper. *GstD1-GFP was* from Fisun Hamaratoglu. *repo-lexA* was from Margaret S Ho. *UAS-Ndi1 and UAS-SoNar* were from Juergen A. Knoblich. *Insc-GAL4* (BDSC:8751) was from Yan Song. *UAS-RedStinger* was BDSC:8546. *UAS-Fer1HCH::GFP; UAS-Fer2LCH::mCherry* was from Fanis Missirlis. *Fer1HCH^{G188}* was from Bing Zhou. Other fly lines used in this study were bought from Bloomington *Drosophila* Stock Center, Tsinghua Fly Center, Vienna *Drosophila* Resource Center, and *Drosophila* Genetic Resource Center (*Supplementary file 1* and *Supplementary file 2*).

## Screen

200 RNAi lines were screened in this study. These genes were related to classical signaling pathways, high expression in glial cells, or secretory protein. *UAS-RNAi* lines were crossed with *repo-Gal4*, and then the third-instar larvae of F1 were dissected to harvest brains. Brains were performed immunostaining with Dpn and PH3. Finally, brains were observed in Confocal Microscope.

## Mouse husbandry

All mouse experiments were performed under the guidelines of the Institutional Animal Care and Use Committee at Southeast University (Approval number: 20211104002). Mice were maintained in a barrier facility, at 25°C, on a regular 12 hr light and 12 hr dark cycle.

## Cell line

GL261-luc murine glioma cell line was purchased from Shanghai Zhong Qiao Xin Zhou Biotechnology Co., Ltd. These cells were identified by STR and tested negative for mycoplasma contamination. Cells were cultured in DMEM medium (Gibco) supplemented with 10% FBS (Gibco) and 1% penicillin/streptomycin (Gibco) in an incubator with 5% $CO_2$ at 37 °C.

## DNA and plasmids

For generating RNAi-resistant UAS-Fer1HCH and UAS-Fer2LCH constructs used for transgenic fly lines, Fer1HCH cDNA (PA type) and Fer2LCH cDNA (PA type) with silent mutations in the region targeted by siRNA were inserted into pUAST plasmids, respectively. The cDNA with silent mutations was synthesized by Sangon Biotech (Shanghai) Co., Ltd. To generate transgenic fly lines of lexAop-Fer1HCH-RNAi and lexAop-Fer2LCH-RNAi, the shRNA sequences targeted on Fer1HCH and Fer2LCH were generated by PCR amplification of UAS-Fer1HCH-RNAi (TH01861.N) and UAS-Fer2LCH-RNAi (THU5585) DNA using primers (5'- ACGGAGCGACAATTCAATTCA-3'/5'-TGATGCCTACCTGATG CCAA-3'), respectively. The fragments were digested by EcoRI and XbaI, and then ligated into lexAop vectors. The constructs were verified by sequencing.

## Immunostaining and antibodies

Larval brains were dissected in PBS (1x), fixed in 4% paraformaldehyde/PBS for 20 min, and washed three times with 0.3% PBST (Triton X-100). Then the brains were blocked in 1% BSA for 1 hr at room temperature. Finally, these brains were incubated with antibodies overnight at 4°C. Primary antibodies: Guinea pig anti-Dpn (1:1000, gift from Juergen A. Knoblich), rabbit anti-PH3 (1/100, CST, 9701), rat anti-Mira (1/1000, abcam, ab197788), mouse anti-Repo (1/100, DSHB, 8D12), rabbit anti-GFP (1/1000, Torrey Pines Biologies, TP401), rabbit anti-Caspase3 (1/100, CST, 9661), mouse anti-Pros (1/20, DSHB), mouse anti-aPKC (1/50, Santa Cruz, Sc-17781).

To detect primary antibodies, the following Alexa-Flour conjugated second antibodies from Invitrogen were used: goat anti-rat Alexa 555 (1/100, A21434), goat anti-guinea pig Alexa 555 (1/100, A21435), donkey anti-rabbit Alexa 488 (1/100, A21206), donkey anti-rat Alexa 488 (1/100, A21208), donkey anti-mouse Alexa 488 (1/100, A21202), goat anti-rat Alexa 633 (1/100, A21094), donkey anti-mouse Alexa 647 (1/100, A31571). All immunostaining images were captured using an LSM700 or an LSM900 (Zeiss) confocal microscope.

## EdU and TUNEL

Dissected freshly brains were incubated in EdU solution for about 30 min to incorporate EdU, then fixed and permeabilized referring to the steps from immunostaining. EdU was detected by Click-iT EdU Alexa Fluor 555 Imaging Kit according to the instructions. TUNEL detection was determined using a kit based on the manufacturer's instructions.

## RNA isolation, semiquantitative RT-PCR, and quantitative real-time PCR

Total RNA was extracted from larval brains by a TRIzol reagent. cDNA was reverse-transcribed from total RNA using a kit. Semiquantitative RT-PCR was performed using the specific primers to amplify the targeted gene segment. Quantitative Real-time PCR was performed on LightCycler 96 (Bio-Rad) using qPCR SuperMix (TransGen Biotech). Following primers were used:

    rp49 F: GCACCAAGCACTTCATCC
    rp49 R: CGATCTCGCCGCAGTAAA
    Fer1HCH F: ATGGTGAAACTAATTGCTAGC
    Fer1HCH R: TCAGATCGCTGACTCCCTC
    Fer2LCH F: GCATGCTCTACGTCAGCCT
    Fer2LCH R: TTACTGCTTCTGCAGATACTC
    ND15 F: ATGTCGCTTACCCCCTTTCTACG
    ND15 R: CTAGTAGGCATCAACGCGTGG
    CG15715 F: ATGGCACGTGGACACCAG
    CG15715 R: TCAGACCTCCTTCAGCTCCTCG

## Western blot and antibodies

Brains were dissected in PBS, and then homogenized in the RIPA buffer with SDS. Homogenate was centrifuged at 12,000 rpm for 5 min at 4°C three times and boiled for 10 min. The protein was separated on 15% SDS-PAGE and transferred to nitrocellulose membranes. The membranes were blocked in 5% milk and incubated overnight at 4°C with antibodies. The blots were detected using Tanon 5200. Primary antibodies used: rabbit anti-Fer1HCH (1/500, this study), rabbit anti-Fer2LCH (1/500, this study), mouse anti-tubulin (1/1000, abcam, ab7291), mouse anti-Repo (1/200, DSHB, 8D12), rabbit anti-4-HNE (1/1000, abcam, ab46545). Rabbit polyclonal antibodies were generated against recombinant proteins His-Fer1HCH or His-Fer2LCH. In brief, the aa 23–200 of Fer1HCH (PA type) or aa 41–227 of Fer2LCH (PA type), contained by different Fer1HCH or Fer2LCH cDNA isoforms, was synthesized and cloned into pET28a vector, respectively. The recombinant proteins were expressed in *E. coli*, extracted, and injected into rabbits for generating the antibodies (Abclonal Biotech.). Then the antibodies were affinity-purified and applied in western blot.

## Temperature shift

GAL80$^{ts}$ was introduced into this system to control the function of GAL4 according to the temperature for determining the temporal window during which ferritin functions to maintain NBs. The eggs of *tubulin-GAL80$^{ts}$; repo-gal4* crossing with *UAS-Fer1HCH RNAi* or *UAS-Fer2LCH RNAi* were collected, and cultured at 18°C (permissive temperature for GAL80$^{ts}$) before hatching, then transferred to 29°C (restrictive temperature for GAL80$^{ts}$) after hatching, or vice versa until third-instar larval stage. Then brains were observed and analyzed.

## Culture medium

Supplemented iron salt and iron chelators (1x) in *Drosophila* food were used as follows: 5 mM ferric ammonium citrate (FAC) (sigma), 0.1 mM bathophenanthrolinedisulfonic acid disodium (BPS) (sigma), 2.5 mM deferiprone (DFP) (sigma).

## Cell dissociation and FACS

NB lineages were labeled with DsRed drive by *Insc-GAL4*. These cells were dissociated and sorted according to the previous protocol (*Harzer et al., 2013*). Briefly, brains were dissected in Schneider's medium and transferred into Rinaldini's solution. Then Rinaldini's solution was removed, and dissociation solution was added to the brains to incubate at 30°C for 1 hr. After washing these brains twice in Rinaldini's solution, the brains were disrupted in PBS. Finally, NB lineages were sorted by flow cytometer (Beckman) based on the red signal.

## RNA sequencing and data processing

### Smart-Seq2 libraries preparation

The sorted NBs were collected into microtubes with ribonuclease inhibitors and lysis components. An Oligo-dT primer was added to the reverse transcription reaction for first-strand cDNA synthesis, followed by PCR amplification to enrich the cDNA and magbeads purification to clean up the production. Then qualified cDNA was sheared randomly by ultrasonic waves for Illumina library preparation protocol including DNA fragmentation, end repair, 3' ends A-tailing, adapter ligation, PCR amplification, and library validation. After library preparation, PerkinElmer LabChipGX Touch and Step OnePlus Real-Time PCR System were introduced for library quality inspection. Qualified libraries were then loaded on Illumina Hiseq platform for PE150 sequencing (Annoroad Gene Technology Co., Ltd).

### Quantification and differential expression analysis

The reads containing adapter sequences, low-quality reads, or undetermined bases were removed from raw data using the fastp program. The clean reads were aligned to the reference genome of *Drosophila melanogaster* using HISAT2 (v2.1.0). The Ensembl database was chosen as the annotation reference. The gene expression was calculated as Fragments Per Kilobase Million mapped reads (FPKM). The differential expression of genes was analyzed with the DEGSeq package. The criterion of |log2 (fold-change)|≥1 and p-value <0.05 was used to identify differentially expressed genes.

## GO and KEGG pathway enrichment

Gene Ontology (GO) enrichment analysis and Kyoto Encyclopedia of Genes and Genomes (KEGG) pathway analysis were performed to investigate the function of differentially expressed genes. GO analysis was used to explore the function of genes, such as biological processes, cellular components, and molecular functions. KEGG analysis focuses on finding and analyzing interactions in biological systems, such as signal transduction and disease pathways. Only the gene sets with p-value <0.05 were significantly enriched.

## Aconitase activity assay

Protein was extracted with assay buffer from Aconitase Activity Assay Kit (sigma, MAK051). The concentration of protein was determined by Pierce Rapid Gold BCA Protein Assay Kit (Thermo Fisher, A53225). Aconitase was measured according to the Aconitase Activity Assay Kit.

## TEM observation and analysis

TEM samples were prepared by standard procedures (*Guangming et al., 2020*). In brief, brains from third instar larvae upon WT and glial ferritin knockdown were dissected in fresh PBS and fixed at 4 °C overnight in a mixture of 2% glutaraldehyde and 2% formaldehyde in 0.1 M sodium cacodylate buffer (pH 7.4), followed by several rinses with cacodylate buffer. The samples were then postfixed for 2 hr with 1% $OsO_4$ in cacodylate buffer and rinsed twice with distilled water. The preparations were stained for 2 hr with 2% saturated uranyl acetate in distilled water and rinsed twice with distilled water. The specimens were dehydrated in an ethanol series, passed through propylene oxide two times, and embedded into a sheet in Epon812 (SPI Science). Each slice was 90 nm thick, and 30–40 slices were gathered into a group and attached to a grid. Samples were analyzed with a Hitachi H-7650 electron microscope operated at 80kV.

## Orthotopic glioma model and DFP injection

To establish an orthotopic glioma model (*Wang et al., 2022*), C57BL/6 mice (4–6 weeks) were anesthetized using 2% isoflurane and positioned in a stereotactic instrument. GL261-luc cell suspension ($1×10^5$ cells in 3 μL PBS) was injected into the striatum. Specifically, the injection site is at 0.5 mm anterior, 2 mm lateral to the bregma, and 3.5 mm below the skull. The injection was done slowly about 10 min.

The mice with glioma were injected intraperitoneally with DFP every 48 hr (*Eybl et al., 2002*). DFP was dissolved in saline (0.9% NaCl) to 10 mg/ml. The doses of injection in mice were 150 mg/kg using 10 mg/ml DFP. The injection volume was determined based on murine weight.

## Bioluminescence imaging

For tumor imaging, mice were injected with luciferin (150 μg/g) in PBS (*Wang et al., 2019*). After 10 min, the mice were imaged using the IVIS Spectrum (PerkinElmer).

## HE staining

At the time of sacrifice, mouse brains were removed and fixed in 4% paraformaldehyde for 24 hr. After fixation, brains were immersed in 30% sucrose and embedded in OCT for cryosectioning. Sections were stained by Hematoxylin and Eosin (HE). Tumor volumes were estimated using Image J.

## Quantifications and statistical analysis

For quantification of NSCs, Dpn- or Mira-positive NSC of CB or the thoracic VNC at the indicated stage were counted. Mitotic index is the number of PH3-positive cells among Dpn-positive cells. For tumor size in *Drosophila*, the area of CNS was measured by NIH ImageJ. For quantifications of NB size, fluorescence, or western blot intensity, NIH ImageJ was used to measure.

All experiments were repeated at least three times. The figure legends showed the number of samples, statistical parameters, and significance. GraphPad Prism 6 was used to perform statistical analyses. For comparisons between two groups, an unpaired two-sided Student's t-test was employed to assess statistical significance. For comparisons between three of more groups, statistical significance was determined using one-way ANOVA with a Bonferroni test. The log-rank test was used to compare the survival curves of mice. The sample distributions were assumed to be

normal and no particular statistical tests were used to check for normality within the samples. Statistical results were presented as means ± SD and p-values of less than 0.05 were considered statistically significant. Asterisks indicate critical levels of significance ($*p<0.05$; $**p<0.01$; $***p<0.001$; $****p<0.0001$).

## Acknowledgements

We are grateful to Drs. Bing Zhou, Margaret S Ho, Juergen A Knoblich, Fanis Missirlis, Fisun Hamaratoglu, Yan Song, Andrea H Brand; Bloomington *Drosophila* Stock Center (Indiana University, Bloomington, Indiana), the *Drosophila* Genetic Resource Center (Kyoto Institute of Technology, Kyoto, Japan), the Vienna *Drosophila* RNAi Center (Vienna, Austria), and Tsinghua Fly Center (Tsinghua University) for fly stocks and reagents. We thank Dr. Bing Zhou for the useful advice on this study. We thank Drs. Junhua Geng, and Mingdao Mu for comments on the manuscript. National Natural Science Foundation of China (grant no. 32100784, grant recipient, Menglong Rui; grant no. 31970675, grant recipient, Su Wang).

## Additional information

### Funding

| Funder | Grant reference number | Author |
| --- | --- | --- |
| National Natural Science Foundation of China | 32100784 | Menglong Rui |
| National Natural Science Foundation of China | 31970675 | Su Wang |

The funders had no role in study design, data collection, and interpretation, or the decision to submit the work for publication.

### Author contributions

Zhixin Ma, Resources, Data curation, Formal analysis, Validation, Investigation, Methodology, Writing - original draft; Wenshu Wang, Resources, Formal analysis, Investigation, Methodology; Xiaojing Yang, Formal analysis, Methodology; Menglong Rui, Funding acquisition, Writing - review and editing; Su Wang, Conceptualization, Resources, Supervision, Funding acquisition, Investigation, Project administration, Writing - review and editing

### Author ORCIDs

Zhixin Ma https://orcid.org/0000-0002-9802-4478
Su Wang https://orcid.org/0000-0002-3167-6420

### Ethics

All mice experiments were performed under the guidelines of the Institutional Animal Care and Use Committee at Southeast University (Approval number: 20211104002).

Reviewer #1 (Public Review): https://doi.org/10.7554/eLife.93604.3.sa1
Reviewer #2 (Public Review): https://doi.org/10.7554/eLife.93604.3.sa2
Author response https://doi.org/10.7554/eLife.93604.3.sa3

## Additional files

### Supplementary files

• Supplementary file 1. Key resources table.
• Supplementary file 2. The phenotype of iron-related genes driven by *repo-GAL4* or *Insc-GAL4*.
• MDAR checklist

## Data availability

Sequencing data have been deposited in GEO under accession codes GSE237124. The raw data has been included as supplements to the corresponding figures. Data from all experiments has been deposited at Dryad.

The following datasets were generated:

| Author(s) | Year | Dataset title | Dataset URL | Database and Identifier |
|---|---|---|---|---|
| Ma Z, Yang X, Wang S | 2024 | Glial ferritin maintains self-renewal and proliferation of neural stem cells by supplying iron in *Drosophila* | https://www.ncbi.nlm.nih.gov/geo/query/acc.cgi?acc=GSE237124 | NCBI Gene Expression Omnibus, GSE237124 |
| Ma Z, Wang W, Yang X, Rui M, Wang S | 2024 | Glial ferritin maintains neural stem cells via transporting iron required for self-renewal in *Drosophila* | https://doi.org/10.5061/dryad.b5mkkwhnq | Dryad Digital Repository, 10.5061/dryad.b5mkkwhnq |

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
