## [Editor Report · eLife assessment]

This **valuable** study, which seeks to identify factors from the glial niche that support and maintain neural stem cells, reports a novel role for ferritin in this process. The authors provide **solid** evidence that defects in larval brain development in Drosophila, resulting from ferritin knockdown, can be attributed to impaired Fe-S cluster activity and ATP production. The findings of this well-conducted study will be of interest to oncologists and neurobiologists.

---

## [Referee Report · Reviewer #1 (Public Review)]

This study unveils a novel role for ferritin in Drosophila larval brain development. Furthermore, it pinpoints that the observed defects in larval brain development resulting from ferritin knockdown are attributed to impaired Fe-S cluster activity and ATP production. Overall this is a well-conducted and novel study.

The author have adequately addressed the concerns.

---

## [Referee Report · Reviewer #2 (Public Review)]

Summary:

Zhixin and collaborators have investigated if the molecular pathways present in glia play a role in the proliferation, maintenance and differentiation of Neural Stem Cells. In this case, Drosophila Neuroblasts are used as models. Authors find that neuronal iron metabolism modulated by glial ferritin is an essential element for Neuroblast proliferation and differentiation. They show that loss of glial ferritin is sufficient to impact the number of neuroblasts. Remarkably, authors have identified that ferritin produced in the glia is secreted to be used as an iron source by the neurons. Therefore iron defects in glia have serious consequences in neuroblasts and likely vice versa. Interestingly, preventing iron absorption in the intestine is sufficient to reduce NB number. Furthermore, they have identified Zip13 as another regulator of the process. Evidence presented strongly indicates that the loss of neuroblasts is due to premature differentiation rather than cell death.

Strengths:

- Comprehensive analysis of the impact of glial iron metabolism in neuroblast behaviour by genetic and drug-based approaches as well as using a second model (mouse) for some validations.

- Using cutting edge methods such as RNAseq as well as very elegant and clean approaches such as RNAi-resistant lines or temperature-sensitive tools

- Goes beyond the state of the art highlighting iron as a key element in neuroblast formation as well as as a target in tumor treatments.

Comments on latest version:

The authors have successfully and convincingly addressed all comments from this reviewer. The modifications, changes and additions have increased the robustness of the results and clearly increased the readability of the manuscript.

This reviewer also appreciates all the efforts and extra work conducted by the authors to finish in a reasonable time all the experiments suggested by all reviewers.

---

## [Author Response]

The following is the authors’ response to the original reviews.

**Reviewer #3 (Public Review):**
The iron manipulation experiments are in the whole animal and it is likely that this affects general feeding behaviour, which is known to affect NB exit from quiescence and proliferative capacity. The loss of ferritin in the gut and iron chelators enhancing the NB phenotype are used as evidence that glia provide iron to NB to support their number and proliferation. Since the loss of NB is a phenotype that could result from many possible underlying causes (including low nutrition), this specific conclusion is one of many possibilities.

We have investigated the feeding behavior of fly by Brilliant Blue (sigma, 861146)[1]. Our result showed that the amount of dye in the fly body were similar between control group and BPS group, suggesting that BPS almost did not affect the feeding behavior (Figure 3—figure supplement 1A).

**Recommendations for the authors:**
Reviewer #1 (Recommendations For The Authors):There was a gap between the Pros nuclear localization and downstream targets of ferritin, particularly NADH dehydrogenase and biosynthesis. Could overexpression of Ndi1 restore Pros localization in NBs?

Ferritin defect downregulates iron level, which leads to cell cycle arrest of NBs via ATP shortage. And cell cycle arrest of NBs probably results in NB differentiation[2, 3]. We have added the experiment in Figure 5—figure supplement 2. This result showed that overexpression of Ndi1 could significantly restore Pros localization in NBs.

The abstract requires revision to cover the major findings of the manuscript, particularly the second half.

We revised the abstract to add more major findings of the manuscript in the second half as follows:

“Abstract

Stem cell niche is critical for regulating the behavior of stem cells. Drosophila neural stem cells (Neuroblasts, NBs) are encased by glial niche cells closely, but it still remains unclear whether glial niche cells can regulate the self-renewal and differentiation of NBs. Here we show that ferritin produced by glia, cooperates with Zip13 to transport iron into NBs for the energy production, which is essential to the self-renewal and proliferation of NBs. The knockdown of glial ferritin encoding genes causes energy shortage in NBs via downregulating aconitase activity and NAD+ level, which leads to the low proliferation and premature differentiation of NBs mediated by Prospero entering nuclei. More importantly, ferritin is a potential target for tumor suppression. In addition, the level of glial ferritin production is affected by the status of NBs, establishing a bicellular iron homeostasis. In this study, we demonstrate that glial cells are indispensable to maintain the self-renewal of NBs, unveiling a novel role of the NB glial niche during brain development.”

In Figure 2B Mira appeared to be nuclear in NBs, which is inconsistent with its normal localization. Was it Dpn by mistake?

In Figure 2B, we confirmed that it is Mira. Moreover, we also provide a magnified picture in Figure 2B’, showing that the Mira mainly localizes to the cortex or in the cytoplasm as previously reported.

Figure 2C, Fer1HCH-GFP/mCherry localization was non-uniform in the NBs revealing 1-2 regions devoid of protein localization potentially corresponding to the nucleus and Mira crescent enrichment. It is important to co-label the nucleus in these cells and discuss the intracellular localization pattern of Ferritin.

We have revised the picture with nuclear marker DAPI in Figure 2C. The result showed that Fer1HCH-GFP/Fer2LCH-mCherry was not co-localized with DAPI, which indicated that *Drosophila* ferritin predominantly distributes in the cytosol[4, 5]. As for the concern mentioned by this reviewer, GFP/mCherry signal in NBs was from glial overexpressed ferritin, which probably resulted in non-uniform signal.

In Figure 3-figure supplement 3F, glial cells in Fer1HCH RNAi appeared to be smaller in size. This should be quantified. Given the significance of ferritin in cortex glial cells, examining the morphology of cortex glial cells is essential.

In Figure 3—figure supplement 3F, we did not label single glial cells so it was difficult to determine whether the size was changed. However, it seems that the chamber formed by the cellular processes of glial cells becomes smaller in *Fer1HCH RNAi*. The glial chamber will undergo remodeling during neurogenesis, which responses to NB signal to enclose the NB and its progeny[6]. Thus, the size of glial chamber is regulated by NB lineage size. In our study, ferritin defect leads to the low proliferation, inducing the smaller lineage of each NB, which likely makes the chamber smaller.

Since the authors showed that the reduced NB number was not due to apoptosis, a time-course experiment for glial ferritin KD is recommended to identify the earliest stage when the phenotype in NB number /proliferation manifests during larval brain development.

We observed brains at different larval stages upon glial ferritin KD. The result showed that NB proliferation decreased significantly, but NB number declined slightly at the second-instar larval stage (Figure 5—figure supplement 1E and F), suggesting that brain defect of glial ferritin KD manifests at the second-instar larval stage.

Transcriptome analysis on ferritin glial KD identified genes in mitochondrial functions, while the in vivo EM data suggested no defects in mitochondria morphology. A short discussion on the inconsistency is required.

For the observation of mitochondria morphology via the in vivo EM data, we focused on visible cristae in mitochondria, which was used to determine whether the ferroptosis happens[7]. It is possible that other details of mitochondria morphology were changed, but we did not focus on that. To describe this result more accurately, we replaced “However, our observation revealed no discernible defects in the mitochondria of NBs after glial ferritin knockdown” with the “However, our result showed that the mitochondrial double membrane and cristae were clearly visible whether in the control group or glial ferritin knockdown group, which suggested that ferroptosis was not the main cause of NB loss upon glial ferritin knockdown” in line 207-209.

The statement “we found no obvious defects of brain at the first-instar larval stage (0-4 hours after larval hatching) when knocking down glial ferritin (Figure 5-figure supplement 1C).” lacks quantification of NB number and proliferation, making it challenging to conclude.

We have provided the quantification of NB number and proliferation rate of the first-instar larval stage in Figure 5—figure supplement 1C and D. The data showed that there is no significant change in NB number and proliferation rate when knocking down ferritin, suggesting that no brain defect manifests at the first-instar larval stage.

A wild-type control is necessary for Figure 6A-C as a reference for normal brain sizes.

We have added *Insc>mCherry RNAi* as a reference in Figure 6A-D, which showed that the brain size of tumor model is larger than normal brain. Moreover, we removed *brat RNAi* data from Figure 6A-D to Figure 6—figure supplement 1A-D for the better layout.

In Figures 6B, D, “Tumor size” should be corrected to “Larval brain volume”.

Here, we measured the brain area to assess the severity of the tumor via ImageJ instead of 3D data of the brain volume. So we think it would be more appropriate to use the “Larval brain size” than “Larval brain volume” here. Thus, we have corrected “Tumor size” to “Larval brain size” in Figure 6B and D to Figure 6—figure supplement 1B and D.

Considering that asymmetric division defects in NBs may lead to premature differentiation, it is advisable to explore the potential involvement of ferritin in asymmetric division.

aPKC is a classic marker to determine the asymmetric division defect of NB. We performed the aPKC staining and found it displayed a crescent at the apical cortex based on the daughter cell position whether in control or glial ferritin knockdown (Figure 5—figure supplement 3A). This result indicated that there was no obvious asymmetric defect after glial ferritin knockdown.

In the statement "Secondly, we examined the apoptosis in glial cells via Caspase-3 or TUNEL staining, and found the apoptotic signal remained unchanged after glial ferritin knockdown (Figure 3-figure supplement 3A-D).", replace "the apoptosis in glial cells" with "the apoptosis in larval brain cells".

We have replaced "the apoptosis in glial cells" with "the apoptosis in larval brain cells" in line 216.

Include a discussion on the involvement of ferritin in mammalian brain development and address the limitations associated with considering ferritin as a potential target for tumor suppression.

We have added the discussion about ferritin in mammalian brain development in line 428-430 and limitation of ferritin for suppressing tumor in line 441-444.

Indicate Insc-GAL4 as BDSC#8751, even if obtained from another source. Additionally, provide information on the extensively used DeRed fly stock used in this study within the methods section.

We provided the stock information of *Insc-GAL4* and *DsRed* in line 673-674.

**Reviewer #2 (Recommendations For The Authors):**
Major points:The number of NBs differs a lot between experiments. For example, in Fig 1B and 1K controls present less than 100 NBs whereas in Figure 1 Supplementary 2B it can be seen that controls have more than 150. Then, depending on which control you compare the number of NBs in flies silencing Fer1HCH or Fer2LCH, the results might change. The authors should explain this.

Figure 1 Supplementary 2B (Figure 1 Supplementary 3B in the revised version) shows NB number in VNC region while Fig 1B and 1K show NB number in CB region. At first, we described the general phenotype showing the NB number in CB and VNC respectively (Fig 1 and Fig 1-Supplementary 1 and 3 in the revised version). And the NB number is consistent in each region. After then, we focused on NB number in CB for the convenience.

This reviewer encourages the authors to use better Gal4 lines to describe the expression patterns of ferritins and Zip13 in the developing brain. On the one hand, the authors do not state which lines they are using (including supplementary table). On the other hand, new Trojan GAL4 (or at least InSite GAL4) lines are a much better tool than classic enhancer trap lines. The authors should perform this experiment.

All stock source and number were documented in Table 2. Ferritin GAL4 and Zip13 GAL4 in this study are InSite GAL4. In addition, we also used another Fer2LCH enhancer trapped GAL4 to verify our result (DGRC104255) and provided the result in Figure 2—figure supplement 1. Our data showed that DsRed driven by *Fer2LCH-GAL4* was co-localized with the glia nuclear protein Repo, instead of the NB nuclear protein Dpn, which was consistent with the result of Fer1HCH/Fer2LCH GAL4. In addition, we will try to obtain the Trojan GAL4 (Fer1HCH/Fer2LCH GAL4 and Zip13 GAL4) and validate this result in the future.

The authors exclude very rapidly the possibility of ferroptosis based only on some mitochondrial morphological features without analysing the other hallmarks of this iron-driven cell death. The authors should at least measure Lipid Peroxidation levels in their experimental scenario either by a kit to quantify by-products of lipid peroxidation such as Malonaldehide (MDA) or using an anti 4-HNE antibody.

We combined multiple experiments to exclude the possibility of ferroptosis. Firstly, ferroptosis can be terminated by iron chelator. And we fed fly with iron chelator upon glial ferritin knockdown, but NB number and proliferation were not restored, which suggested that ferroptosis probably was not the cause of NB loss induced by glial ferritin knockdown (Figure 3B and C). Secondly, Zip13 transports iron into the secretary pathway and further out of the cells in Drosophila gut[8]. Our data showed that knocking down iron transporter Zip13 in glia resulted in the decline of NB number and proliferation, which was consistent with the phenotype upon glial ferritin knockdown (Figure 3E-G). More importantly, the knockdown of Zip13 and ferritin simultaneously aggravated the phenotype in NB number and proliferation (Figure 3E-G). These results suggested that the phenotype was induced by iron deficiency in NB, which excluded the possibility of iron overload or ferroptosis to be the main cause of NB loss upon glial ferritin knockdown. Finally, we observed mitochondrial morphology on double membrane and the cristae that are critical hallmarks of ferroptosis, but found no significant damage (Figure 3-figure supplement 2E and F).

In addition, we have added the 4-HNE determination in Figure 3—figure supplement 2G and H. This result showed that 4-HNE level did not change significantly, suggesting that lipid peroxidation was stable, which supported to exclude the possibility that the ferroptosis led to the NB loss upon glial ferritin knockdown.

All of the above results together indicate that ferroptosis is not the cause of NB loss after ferritin knockdown.

A major flaw of the manuscript is related to the chapter Glial ferritin defects result in impaired Fe-S cluster activity and ATP production and the results displayed in Figure 4. The authors talk about the importance of FeS clusters for energy production in the mitochondria. Surprisingly, the authors do not analyse the genes involved in this process such as but they present the interaction with the cytosolic FeS machinery that has a role in some extramitochondrial proteins but no role in the synthesis of FeS clusters incorporated in the enzymes of the TCA cycle and the respiratory chain. The authors should repeat the experiments incorporating the genes NSF1 (CG12264), ISCU(CG9836), ISD11 (CG3717), and fh (CG8971) or remove (or at least rewrite) this entire section.

Thanks for this constructive advice and we have revised this in Figure 4B and C. We repeated the experiment with blocking mitochondrial Fe-S cluster biosynthesis by knocking down *Nfs1 (CG12264)*, *ISCU(CG9836), ISD11 (CG3717),* and *fh (CG8971)*, respectively. *Nfs1* knockdown in NB led to a low proliferation, which was consistent with CIA knockdown. However, we did not observe the obvious brain defect in *ISCU(CG9836), ISD11 (CG3717),* and *fh (CG8971)* knockdown in NB. Our interpretation of these results is that Nfs1 probably is a necessary core component in Fe-S cluster assembly while others are dispensable[9].

The presence and aim of the mouse model is unclear to this reviewer. On the one hand, it is not used to corroborate the fly findings regarding iron needs from neuroblasts. On the other hand, and without further explanation, authors migrate from a fly tumor model based on modifying all neuroblasts to a mammalian model based exclusively on a glioma. The authors should clarify those issues.

Although iron transporter probably is different in *Drosophila* and mammal, iron function is conserved as an essential nutrient for cell growth and proliferation from *Drosophila* to mammal. The data of fly suggested that iron is critical for brain tumor growth and thus we verified this in mammalian model. Glioma is the most common form of central nervous system neoplasm that originates from neuroglial stem or progenitor cells[10]. Therefore, we validated the effect of iron chelator DFP on glioma in mice and found that DFP could suppress the glioma growth and further prolong the survival of tumor-bearing mice.

Minor pointsAlthough referred to adult flies, the authors did not include either in the introduction or in the discussion existing literature about expression of ferritins in glia or alterations of iron metabolism in fly glia cells (PMID: 21440626 and 25841783, respectively) or usage of the iron chelator DFP in drosophila (PMID: 23542074). The author should check these manuscripts and consider the possibility of incorporating them into their manuscript.

Thanks for your reminder. We have incorporated all recommended papers into our manuscript line 65-67 and 168.

The number of experiments in each figure is missing.

All experiments were repeated at least three times. And we revised this in Quantifications and Statistical Analysis of Materials and methods.

If graphs are expressed as mean +/- sem, it is difficult to understand the significance stated by the authors in Figure 2E.

We apologize for this mistake and have revised this in Quantifications and Statistical Analysis. All statistical results were presented as means ± SD.

When authors measure aconitase activity, are they measuring all (cytosolic and mitochondrial) or only one of them? This is important to better understand the experiments done by the authors to describe any mitochondrial contribution (see above in major points).

In this experiment, we were measuring the total aconitase activity. We also tried to determine mitochondrial aconitase but it failed, which was possibly ascribed to low biomass of tissue sample.

In this line, why do controls in aconitase and atp lack an error bar? Are the statistical tests applied the correct ones? It is not the same to have paired or unpaired observations.

It is the normalization. We repeated these experiments at least three times in different weeks respectively, because the whole process was time-consuming and energy-consuming including the collection of brains, protein determination and ATP or aconitase determination. And the efficiency of aconitase or ATP kit changed with time. We cannot control the experiment condition identically in different batches. Therefore, we performed normalization every time to present the more accurate result. The control group was normalized as 1 via dividing into itself and other groups were divided by the control. This normalized process was repeated three times. Therefore, there is no error bar in the control group. We think it is appropriate to apply ANOVA with a Bonferroni test in the three groups.

In some cases, further rescue experiments would be appreciated. For example, expression of Ndi restores control NAD+ levels or number of NBs, it would be interesting to know if this is accompanied by restoring mitochondrial integrity and its ability to produce ATP.

We have determined ATP production after overexpressing Ndi1 and provided this result in Figure 4—figure supplement 1B. The data showed that expression of Ndi1 could restore ATP production upon glial *Fer2LCH* knockdown, which was consistent with our conclusion.

Lines 293-299 on page 7 are difficult to understand.

According to our above results, the decrease of NB number and proliferation upon glial ferritin knockdown (KD) was caused by energy deficiency. As shown in the schematic diagram (Author response image 1), “T” represented the total energy which was used for NB maintenance and proliferation. “N” indicated the energy for maintaining NB number. “P” indicated the energy for NB proliferation. “T” is equal to “N” plus “P”. When ferritin was knocked down in glia, “T”, “N” and “P” declined in “Ferritin KD” compared to “wildtype (WT)”. Knockdown of *pros* can prevent the differentiation of NB, but it cannot supply the energy for NB, which probably results in the rescue of NB number but not proliferation. Specifically, NB number increased significantly in “Ferritin KD Pros KD” compared to “Ferritin KD”, which resulted in consuming more energy for NB maintenance in “Ferritin KD Pros KD”. As shown in the schematic diagram, “T” was not changed between “Ferritin KD Pros KD” and “Ferritin KD”, whereas ”N” was increased in “Ferritin KD Pros KD” compared to “Ferritin KD”. Thus, “P” was decreased, which suggested that less energy was remained for proliferation, leading to the failure of rescue in NB proliferation. It seemed that the level of proliferation in “Ferritin KD Pros KD” was even lower than “Ferritin KD”.

**Author response image 1. sa3fig1:** The schematic diagram of relationship between energy and NB function in different groups. “T” represents total energy for NB maintenance and proliferation. “N” represents the energy for NB maintenance. “P” represents the energy for NB proliferation. T=N+P

Line 601 should indicate that Tables 2 and 3 are part of the supplementary material.

We have revised this in line 678.

Figure 4-supplement 1. Only validation of 2 genes from a RNAseq seems too little.

We dissected hundreds of brains for sorting NBs because of low biomass of fly brain. This is a difficult and energy-consuming work. Most NBs were used for RNA-seq, so we can only use a small amount of sample left for validation which is not enough for more genes.

Figure 6E, the authors indicate that 10 mg/ml DFP injection could significantly prolong the survival time. Which increase in % is produced by DFP?

We have provided the bar graph in Author response image 2. The increase is about 16.67% by DFP injection.

**Author response image 2. sa3fig2:** The bar graph of survival time of mice treated with DFP. (The unpaired two-sided Student’s t test was employed to assess statistical significance. Statistical results were presented as means ± SD. n=7,6; *: p<0.05)

**Reviewer #3 (Recommendations For The Authors):**
As I read the initial results that built the story (glia make ferritin>release it> NBs take them up>use it for TCA and ETC) I kept thinking about what it meant for NBs to be 'lost'. This led me to consider alternate possibilities that the results might point to, other than the ones the authors were suggesting. It was only in Figure 5 that the authors ruled out some of those possibilities. I would suggest that they first illustrate how NBs are lost upon glial ferritin loss of function before they delve into the mechanism. This would also be a place to similarly address that glial numbers and general morphology are unchanged upon ferritin loss.

This recommendation provides a valuable guideline to build this story especially for researchers who are interested in neural stem cell studies. Actually, we tried this logic to present our study but found that there are several gaps in the middle of the manuscript, such as the relationship between glial ferritin and Pros localization in NB, so that the whole story cannot be fluently presented. Therefore, we decided to present this study in the current way.

More details of the screen would be useful to know. How many lines did they screen, what was the assay? This is not mentioned anywhere in the text.

We have added this in Screen of Materials and methods. We screened about 200 lines which are components of classical signaling pathways, highly expressed genes in glial cells or secretory protein encoding genes. *UAS-RNAi* lines were crossed with *repo-Gal4*, and then third-instar larvae of F1 were dissected. We got the brains from F1 larvae and performed immunostaining with Dpn and PH3. Finally, we observed the brain in Confocal Microscope.

Many graphs seem to be repeated in the main figures and the supplementary data. This is unnecessary, or at least should be mentioned.

We appreciate your kind reminder. However, we carefully went through all the figures and did not find the repeated graphs, though some of them look similar.

The authors mention that they tested which glial subtypes ferritin is needed in, but don't show the data. Could they please show the data? Same with the other iron transport/storage/regulation. Also, in both this and later sections, the authors could mention which Gal4 was used to label what cell types. The assumption is that the reader will know this information.

We have added the result of ferritin knockdown in glial subpopulations in Figure 1—figure supplement 2. However, considering that the quantity of iron-related genes, we did not take the picture, but we recorded this in Table 3.

For all their images showing colocalisation, magnified, single-colour images shown in grayscale will be useful. For example, without the magnification, it is not possible to see the NB expression of the protein trap line in Figure 2B. A magnified crop of a few NBs (not a single one like in 2C) would be more useful.

We have provided Figure 2A’, B’, D’ and Figure 3D’ as suggested.

There are a lot of very specific assays used to detect ROS, NAD, aconitase activity, among others. It would be nice to have a brief but clear description of how they work in the main text. I found myself having to refer to other sources to understand them. (I believe SoNAR should be attributed to Zhao et al 206 and not Bonnay et al 2020.)

We have added a brief description about ROS, aconitase activity, NAD in line 198-199, 229-231, and 269 as suggested.

I did not understand the normalisation done with respect to SoNAR. Is this standard practice? Is the assumption that 'overall protein levels will be higher in slowly proliferating NBs' reasonable? This is why they state the need to normalise.

The SoNAR normalization is not a standard practice. However, we think that our normalization of SoNar is reasonable. According to our results, the expression level of Dpn and Mira seemed higher in glial ferritin knockdown, so we speculated that some proteins accumulated in slowly proliferating NBs. Thus, we used *Insc-GAL4* to drive DsRed for indicating the expression level of Insc and found that DsRed rose after glial ferritin knockdown, suggesting that Insc expression was increased indeed. Therefore, we have to normalize SoNar driven by *Insc-GAL4* based on DsRed driven by *Insc-Gal4*, which eliminates the effect of increased Insc upon glial ferritin knockdown.

FAC is mentioned as a chelator? But the authors seem to use it oppositely. Is there an error?

FAC is a type of iron salt, which is used to supply iron. We have also indicated that in line 156 according to your advice.

The lack of any cell death in the L3 brain surprised me. There should be plenty of hemilineages that die, as do many NBs, particularly in the abdominal segments. Is the stain working? Related to this, P35 is not the best method for rescuing cell death. H99 might be a better way to go.

We were also surprised to see this result and repeated this experiment for several times with both negative and positive controls. Moreover, we also used TUNEL to validate this result, which led to the same result. We will try to use H99 to rescue NB loss in the future, because it needs to be integrated and recombined with our current genetic tools.

It would be nice to see the aconitase activity signal as opposed to just the quantification.

This method can only determine the absorbance for indicating aconitase activity, so our result is just the quantification.

Glia are born after NBs are specified. In fact, they arise from NBs (and glioblasts). So, it's unlikely that the knockdown of ferritin in glia can at all affect initial NB specification.

We completely agree with this statement.

The section on tumor suppression seems out of place. The fly data on which the authors base this as an angle to chase is weak. Dividing cells will be impaired if they have inadequate energy production. As a therapeutic, this will affect every cell in the body. I'm not sure that cancer therapeutics is pursuing such broadly acting lines of therapies anymore.

Our data suggested that iron/ferritin is more critical for high proliferative cells. Tumor cells have a high expression of TfR (Transferrin Receptor)[11], which can bind to Transferrin and ferritin[12]. And ferritin specifically targets on the tumor cells[11]. Thus, we think iron/ferritin is extremely essential for tumor cells. If we can find the appropriate dose of iron/ferritin inhibitor, suppressing tumor growth but maintaining normal cell growth, iron/ferritin might be an effective target of tumor treatment.

The feedback from NB to glial ferritin is also weak data. The increased cell numbers (of unknown identity) could well be contributing to the increase in ferritin. I would omit the last two sections from the MS.

In *brat RNAi* and *numb RNAi*, increased cells are NB-like cells, which cannot undergo further differentiation and are not expected to produce ferritin. More importantly, we used Repo (glia marker) as the reference and quantified the ratio of ferritin level to Repo level, which can exclude the possibility that increased glial cells lead to the increase in ferritin.

References

[1] Tanimura T, Isono K, Takamura T, et al. Genetic Dimorphism in the Taste Sensitivity to Trehalose in Drosophila-Melanogaster. J Comp Physiol, 1982,147(4):433-7

[2] Myster DL, Duronio RJ. Cell cycle: To differentiate or not to differentiate? Current Biology, 2000,10(8):R302-R4

[3] Dalton S. Linking the Cell Cycle to Cell Fate Decisions. Trends in Cell Biology, 2015,25(10):592-600

[4] Nichol H, Law JH, Winzerling JJ. Iron metabolism in insects. Annu Rev Entomol, 2002,47:535-59

[5] Pham DQ, Winzerling JJ. Insect ferritins: Typical or atypical? Biochim Biophys Acta, 2010,1800(8):824-33

[6] Speder P, Brand AH. Systemic and local cues drive neural stem cell niche remodelling during neurogenesis in Drosophila. Elife, 2018,7

[7] Mumbauer S, Pascual J, Kolotuev I, et al. Ferritin heavy chain protects the developing wing from reactive oxygen species and ferroptosis. PLoS Genet, 2019,15(9):e1008396

[8] Xiao G, Wan Z, Fan Q, et al. The metal transporter ZIP13 supplies iron into the secretory pathway in Drosophila melanogaster. Elife, 2014,3:e03191

[9] Marelja Z, Leimkühler S, Missirlis F. Iron Sulfur and Molybdenum Cofactor Enzymes Regulate the Life Cycle by Controlling Cell Metabolism. Front Physiol, 2018,9

[10] Morgan LL. The epidemiology of glioma in adults: a "state of the science" review. Neuro-Oncology, 2015,17(4):623-4

[11] Fan K, Cao C, Pan Y, et al. Magnetoferritin nanoparticles for targeting and visualizing tumour tissues. Nat Nanotechnol, 2012,7(7):459-64

[12] Li L, Fang CJ, Ryan JC, et al. Binding and uptake of H-ferritin are mediated by human transferrin receptor-1. Proc Natl Acad Sci U S A, 2010,107(8):3505-10

[13] Tang X, Zhou B. Ferritin is the key to dietary iron absorption and tissue iron detoxification in *Drosophila melanogaster*. FASEB J, 2013,27(1):288-98

[14] Xiao G, Liu ZH, Zhao M, et al. Transferrin 1 Functions in Iron Trafficking and Genetically Interacts with Ferritin in *Drosophila melanogaster*. Cell Rep, 2019,26(3):748-58 e5

[15] Mukherjee C, Kling T, Russo B, et al. Oligodendrocytes Provide Antioxidant Defense Function for Neurons by Secreting Ferritin Heavy Chain. Cell Metab, 2020,32(2):259-72 e10

16] Knoblich JA, Jan LY, Jan YN. Asymmetric Segregation of Numb and Prospero during Cell-Division. Nature, 1995,377(6550):624-7

[17] Zacharioudaki E, Magadi SS, Delidakis C. bHLH-O proteins are crucial for neuroblast self-renewal and mediate Notch-induced overproliferation. Development, 2012,139(7):1258-69

[18] Bello B, Reichert H, Hirth F. The brain tumor gene negatively regulates neural progenitor cell proliferation in the larval central brain of. Development, 2006,133(14):2639-48

[19] Choksi SP, Southall TD, Bossing T, et al. Prospero acts as a binary switch between self-renewal and differentiation in *Drosophila* neural stem cells. Developmental Cell, 2006,11(6):775-89

[20] Spana EP, Doe CQ. The Prospero Transcription Factor Is Asymmetrically Localized to the Cell Cortex during Neuroblast Mitosis in *Drosophila*. Development, 1995,121(10):3187-95